# Calibrating the GAMIL3-1° climate model using a

# derivative-free optimization method

- Wenjun Liang<sup>1,2,3</sup>, Simon Frederick Barnard Tett<sup>3</sup>, Lijuan Li<sup>4</sup>, Coralia Cartis<sup>5</sup>, Danya
- 4 Xu<sup>6</sup>, Wenjie Dong<sup>1,2\*</sup>, Junjie Huang<sup>4,7</sup>
- <sup>1</sup> School of Atmospheric Sciences, Southern Marine Science and Engineering Guangdong
- 6 Laboratory (Zhuhai), Sun Yat-sen University, Zhuhai, 519082, China
- <sup>2</sup> Key Laboratory of Tropical Atmosphere-Ocean System, Ministry of Education, Zhuhai, 519082,
- 8 China

1

2

- 9 <sup>3</sup> School of Geosciences, University of Edinburgh, Edinburgh, United Kingdom
- <sup>4</sup> State Key Laboratory of Earth System Numerical Modeling and Application, Chinese Academy
- 11 of Sciences, Beijing, China
- <sup>5</sup> Mathematical Institute, University of Oxford, Oxford, United Kingdom
- <sup>6</sup> Southern Marine Science and Engineering Guangdong Laboratory (Zhuhai), Zhuhai 519082,
- 14 China
- <sup>7</sup> Anhui Meteorological Information Centre, Hefei, China
- 16 Correspondence to: Wenjie Dong (dongwj3@mail.sysu.edu.cn)
- 17 **Abstract.** Parameterization in climate models often involves parameters that are
- poorly constrained by observations or theoretical understanding alone. Manual tuning
- by experts can be time-consuming, subjective, and prone to underestimating
- uncertainties. Automated tuning methods offer a promising alternative, enabling faster,
- objective improvements in model performance and better uncertainty quantification.
- This study presents an automated parameter-tuning framework that employs a
- derivative-free optimization solver (DFO-LS) to simultaneously perturb and tune
- multiple convection-related and microphysics parameters. The framework explicitly
- accounts for observational and initial condition uncertainties (internal variability) to
- calibrate a 1-degree resolution atmospheric model (GAMIL3). To evaluate its
- performance, two main tuning experiments were conducted, targeting 10 and 20
- parameters, respectively. In addition, three sensitivity experiments tested the effect of
- varying initial parameter values in the 10-parameter case. Both tuning experiments
- achieved a rapid reduction in the cost function. The 10-parameter optimization
- improved model accuracy for 24 of 34 key variables, while expanding to 20 parameters

yielded improvement for 25 variables, though some structural model biases appeared. Ten-year AMIP simulations validated the robustness and stability of the tuning results, showing that the improvements persisted over extended simulations. Additionally, evaluations of the coupled model with optimized parameters showed, compared to the default parameters settings, reduced climate drift, a more stable climate system, and more realistic sea surface temperatures, despite a residual global energy imbalance of 2.0 W/m² (about 1.4 W/m² arising from the intrinsic imbalance of the atmospheric component) and some remaining regional biases. The sensitivity experiments further underscored the efficiency of the tuning algorithm and highlight the importance of expert judgment in selecting initial parameter values. This tuning framework is broadly applicable to other general circulation models (GCMs), supporting comprehensive parameter tuning and advancing model development.

## 1 Introduction

Assessing current and future climate change risks to natural and human systems heavily relies on numerical simulations using advanced climate or Earth System Models (ESMs). In recent decades, significant progress has been made in advancing the major components of the Earth system-such as the atmosphere, ocean, land, and human systems (Prinn 2012; Bogenschutz et al., 2018; Fox-Kemper et al., 2019; Blockley et al., 2020; Blyth et al., 2021)—as well as in developing the coupling techniques required to form fully integrated ESMs (Valcke et al., 2012; Smith et al., 2021; Liu et al., 2023). However, many unresolved issues remain in the development of ESMs, including but not limited to simulation bias in air-sea interactions (Ham et al., 2013; Bellucci et al., 2021; Wei et al., 2021; Meng et al., 2022), the double Intertropical Convergence Zone (ITCZ) problem (Tian et al., 2020), and the coupling of biogeochemical cycles such as the carbon cycle or nutrient cycles with the physical climate system (Erickson et al., 2008). The complexity of the Earth's climate system and the inherent uncertainties in climate models present significant challenges in achieving reliable projections. One of the key sources of uncertainty arises from the representation of unresolved physical processes through parameterizations (Gentine et al., 2021; Jebeile et al., 2023).































Parameterizations are crucial when accounting for processes that occur at unresolved scales or are missing from the model formulation. Parameterizations provide simplified representations of sub-grid processes like cloud convection and turbulence, which cannot be explicitly resolved at scales smaller than the model's grid resolution. For example, processes such as atmospheric radiative transfer and cloud microphysics are too complex to be represented in full detail within ESMs, so parameterizations offer simplified approximations to capture their essential effects. Parameterization often involves parameters whose values are frequently not wellconstrained by either observations or theory alone (Ludovic, 2021), which can directly affect the performance of the model simulation. Consequently, parameter tuning, the process of estimating these uncertain parameters to minimize the discrepancy between specific observations and model results, becomes a critical step in climate model development (Hourdin et al., 2017). Appropriate parameter tuning enhances the accuracy and skill of climate models by optimizing parameter values to better match observations or high-resolution simulations used as calibration targets (Mauritsen et al., 2012; Bhouri et al., 2023). For example, parameter tuning allows adjusting the values of parameters in parameterizations that approximate these unresolved processes like cloud convection, turbulence, etc (Golaz et al., 2013; Zou et al., 2014; Mignot et al., 2021; Xie et al., 2023). By tuning parameter values during the model calibration process, modelers can partly compensate for known structural errors, deficiencies, or missing processes in the underlying model formulation itself (Williamson et al., 2015a; Hourdin et al., 2017; Tett et al., 2017; Schneider et al., 2024). What's more, exploring the range of plausible parameter values through tuning allows quantifying parametric uncertainties and their impacts on model outputs and projections (Jackson et al., 2004; Neelin et al, 2010; Williamson et al., 2013; Tett et al., 2013; Qian et al., 2016). Broadly speaking, parameter tuning methods aim to quickly optimize a cost

function that measures the distance between model simulations and a small collection

Bellprat et al. (2012), Tett et al. (2013), Yang et al. (2013), Zou et al. (2014), Zhang et al. (2015b), and Tett et al. (2017). For instance, in the experiments conducted by Tett et al. (2017) with an atmospheric GCM, 7 and 14 parameters related to the convection, cloud microphysics, and boundary-layer dynamics (Yamazaki et al., 2013) were estimated using variants of the Gauss-Newton algorithm (Tett et al., 2013) to minimize the differences between simulated and observed large-scale, multi-year averaged net radiative fluxes. These optimized parameters were then applied in a coupled GCM. Zhang et al. (2015b) employed an improved downhill simplex method to optimize seven parameters selected from the convection and cloud-fraction parameterization scheme, and reported successful improvement of an atmospheric model's performance. This improved method overcomes the limitations of the traditional downhill simplex method and offers better computational efficiency compared to evolutionary optimization algorithms.

Traditionally, uncertain parameters have been tuned manually through extensive comparisons of model simulations with available observations. This approach is subjective, labor-intensive, computationally expensive, and can lead to underexploration of the parameter space, potentially underestimating uncertainties and leaving model biases unresolved (Allen et al., 2000; Hakkarainen et al., 2012; Hourdin et al. 2017; Hourdin et al., 2023). By contrast, automatic and objective parameter calibration techniques have advanced rapidly due to their efficiency, effectiveness, and wider applicability (Chen et al., 1999; Elkinton et al., 2008; Bardenet et al., 2013; Zhang et al., 2015b). Bardenet et al. (2013) combined surrogate-based ranking and optimization techniques for surrogate-based collaborative tuning, proposing a generic method to incorporate knowledge from previous experiments. This approach can effectively improve upon manual hyperparameter tuning. Zhang et al. (2015b) proposed a "three-step" methodology for parameters tuning. Before the final step of applying the downhill simplex method, they introduced two preliminary steps: determining the model's sensitivity to the parameters and selecting the optimum initial values for those sensitive parameters. By following this process, they were able to automatically and effectively obtain the optimal combination of key parameters in cloud and convective parameterizations.































However, previous studies were either semi-automatic or lacked sufficient observational constraints, such as the net flux at the top of the atmosphere (TOA). Moreover, earlier objective tuning methods that relied on cost functions often overlooked key sources of uncertainty, including observational uncertainty and the internal variability of variables. To address these limitations, we developed a new objective and automatic parameter tuning framework that is more efficient for tuning parameters in GCMs. Compared to previous automatic tuning efforts, this system operates entirely within a Python environment and includes several new optimization algorithms, including Gauss-Newton (Burke et al., 1995; Kim et al., 2008; Tett et al., 2017), the Python Surrogate Optimization Toolbox (pySOT; Regis and Shoemaker, 2012), and the Derivative-Free Optimizer for Least-Squares (DFO-LS; Cartis et al., 2019; Hough et al., 2022). The DFO-LS package is designed to find local solutions to nonlinear least-squares minimization problems without requiring derivatives of the objective function, and has been numerically tested to be particularly effective in finding global optimization solutions. Our framework supports multiple observations and constraints as optimization targets. Additionally, it considers the internal variability of GCMs and integrates sensitivity analysis with the optimization process, making it a more flexible and efficient model tuning system overall. Moreover, systematically and simultaneously perturbing multiple parameters addresses the concern that optimizing a single objective may lead to suboptimal solutions for other objectives and might overlook the global optimum for the overall tuning metric (Qian et al., 2015; Williamson et al., 2015a). We have designed and implemented an automatic workflow to streamline the calibration process, enhancing efficiency. This method and workflow are readily applicable to GCMs, facilitating accelerated model development processes. Using this framework, we tune the latest released version 3 of the Grid-Point Atmospheric Model developed at the State Key Laboratory of Numerical Modeling for Atmospheric Sciences and Geophysical Fluid Dynamics (LASG) in the Institute of Atmospheric Physics (IAP), named GAMIL3 (Li et al., 2020a). This study demonstrates how the tuning framework can automatically and effectively optimize

- model parameters to achieve better performance against observations.
- Our objectives are as follows:
- 1. To assess the performance of the tuning algorithm in the GAMIL3 atmospheric model;
- 2. To investigate the impact of various parameters and initial values on the tuning results;
- 3. To evaluate the performance of the optimized parameters in decadal simulations and long-term coupled model runs.

The paper is organized as follows: Section 2 introduces the proposed automatic framework, the tuning model and experiments, observational data and metrics, and the tuning algorithm. Section 3 presents the evaluation of the tuning results in short- to long-tern simulations, including coupled model runs. This is followed by a discussion in Section 4 and a conclusion in Section 5.

### 2 Methods

## 2.1 The automatic tuning framework

Here we present the automatic tuning framework (Fig. 1) we have developed, which includes, but is not limited to, functions such as model compiling, (re)submitting, parameter tuning, results evaluation, and diagnostics. Specifically, the framework comprises three main processing modules that collectively control the entire system: the model preprocessing module (the lower left panel in Fig. 1), the model optimizing module (the middle panel in Fig. 1), and the model post-processing module (the right panel in Fig. 1).

The preprocessing module prepares various input data for the optimization process, with particular focus on model internal variations and observational uncertainties (Tett et al., 2017), which will be further discussed in a later section. The optimizing module, which uses the DFO-LS optimization method, is the core component of this tuning system and is primarily responsible for updating model parameters and running simulations. In the initialization of DFO-LS, we use the default parameter settings provided by the DFOLS software package, including the specification of the initial trust

region, which is an algorithm parameter that governs the size of the local search area. Any constraints on the simulated variables are also specified at this stage. The initial trust region radius ( $\it rhobeg$ ) is set to 0.18 (normalized to parameter ranges) based on sensitivity tests. This choice ensures that the first iterations explore locally without overstepping physical plausibility, balancing efficient convergence and sufficient sampling of the parameter space (Cartis et al., 2019). In addition, we apply a constraint to a simulated variable using a parameter  $\mu$ , which determines the weighting of the constraint term (1/(2 $\mu$ ); see Supplementary S1). In this study, following Tett et al (2017, 2022), this constraint is applied to the global average TOA net flux. To tightly constrain this variable,  $\mu$  is set to 0.18 which corresponds to a total uncertainty of 0.15 W/m² somewhat higher than the observational error of 0.1 W/m².































The optimization process begins with a parameter perturbation phase, in which K+1 simulations are conducted: one reference simulation using the initial parameter set, and K additional simulations—each perturbing one of the K tunable parameters individually—relative to the reference. These initial simulations establish baseline parameter sensitivities and provide finite-difference gradient estimates for the DFO-LS algorithm. The subsequent optimization phase then iteratively modifies parameter values through trust-region managed steps, where each iteration evaluates candidate points, updates local quadratic models of the cost function, and adjusts parameters based on actual versus predicted improvement ratios until convergence criteria are satisfied. In addition to the initial K+1 simulation runs required to initialize the DFOLS algorithm for a K-parameter case, each iteration typically involves 1-3 additional model simulations, depending on the trust-region management strategy and the progress of the algorithm. The algorithm normally performs one simulation per iteration to evaluate a new candidate parameter set, but may conduct 3 simulations when the local quadratic model requires improvement or when the actual-to-predicted improvement ratio falls below zero (Cartis et al., 2019). Total evaluations include the initial runs plus all subsequent iterations evaluations. The post-processing module receives the output from the optimization module, including the optimized parameters, the sensitivity of variables to the parameters, and the cost function values from different iterations. It

then help us to conducts a comprehensive diagnostic analysis—examining spatial patterns, process-level responses, parameter sensitivities, and multi-variable metrics—to assess the physical credibility of each solution. This structured yet flexible workflow shifts the modeller's role from manual trial-and-error to the management and interpretation of automated explorations, thereby enhancing both the traceability and objectivity of the modeling process.

### 2.2 Observations and parameter selection








To set up our optimization problem, we focus on the large-scale performance of the 217 model and consider the differences between land and ocean, particularly in the tropical 218 region. This region is characterized by distinct air-sea interactions, such as those over 219 the Western Pacific warm pool (Wyrtki, 1975), the Eastern Pacific equatorial cold 220 tongue region (Philander, 1983), and the Indian Ocean Dipole region (Saji et al., 1999). 221 Therefore, following the methods outlined by Tett et al. (2017), we separate the analysis 222 into four regions based on latitude ( $\theta$ , defined as positive northward from the equator): 223 the northern hemispheric extra-tropical region ( $\theta > 30^{\circ}$  N), the tropical region ( $30^{\circ}$  S  $\geq$ 224 225  $\theta \le 30^{\circ}$  N), subdivided into tropical land and ocean, and the southern hemispheric extratropical region ( $\theta < 30^{\circ}$  S). 226 The observational variables used in this study are detailed in Table 1. While most 227 variables are divided into four regions—labeled TROPICSLAND (tropical land: 228 30° S–30° N over land), TROPICSOCEAN (tropical ocean: 30° S–30° N over ocean), 229 NHX (Northern Hemispheric extra-tropics: >30° N), and SHX (Southern 230 Hemispheric extra-tropics: <-30° S)—each with its own target and uncertainty, 231 NETFLUX is averaged over all regions and serves as a global constraint. For the MSLP 232 233 variable, regional mean values are expressed as anomalies relative to the global mean (delta global mean, denoted by the suffix "DGM"), obtained by subtracting the global 234 average from each regional mean. Specifically, the target values for variables T500, 235 RH500, and MSLP are derived from ECMWF Reanalysis v5 data (ERA5; Hersbach et 236 al., 2020); the radiation variables (OLR, OLRC, RSR, RSRC, and NETFLUX) are 237 sourced from Clouds and the Earth's Radiant Energy System (CERES; Wielicki et al., 238

1998); and the Land Air Temperature (LAT) and Land precipitation (Lprecip) data come from the Climatic Research Unit (CRU; Jones et al., 2012; Harris et al., 2017). The uncertainties of the variables are derived from the absolute error among different data sources, which will be discussed further in section 2.4. All targets and uncertainties of the variables in Table 1 are for the year 2011, primarily used for model optimization.

The atmospheric model parameters we calibrated are detailed in Table 2, encompassing selections from deep convection, shallow convection, microphysics, cloud fraction, and turbulence schemes. The selection of these parameters, along with their default values and plausible ranges, is based on expert judgment as recommended by the GAMIL3 developers and corresponds to the model configuration used in CMIP6 experiments. This approach prevents the optimization from exploring unrealistic regions of parameter space. While the plausible ranges are defined as the maximum physically meaningful bounds (e.g., rhcrit: 0.65–0.95), the constraint on the global average TOA net flux ensures it closely matches the observations after tuning. For visualization, all parameters are normalized based on their plausible ranges, with 0 representing the minimum value of the range and 1 representing the maximum one. Then two experiments are conducted to assess the impacts of varying the number of parameters on the optimized results:

- 1. We selected the first 10 parameters (listed in the first column of Table 2) from deep convection, shallow convection, microphysics, and cloud fraction schemes. These parameters are identified as the most sensitive to the model's performance based on Xie et al. (2023), and are therefore chosen for tuning. This case is denoted as the "10-param." case in the captions of all relevant figures.
- 2. An additional set of the next 10 parameters (also listed in the first column of Table 2), related to microphysics and turbulence schemes, is included alongside the initial 10 parameters. This approach aims to explore the impact of varying the number of tuning parameters on the optimization results. This case is denoted as the "20-param." case in the captions of all relevant figures.

## 2.3 Model description and experiments






























In this study, we employ GAMIL3, which adopts a finite difference dynamical core and a weighted equal-area longitude-latitude grid to maintain numerical stability near the polars without the need for filtering or smoothing (Wang et al., 2004; Li et al., 2020a). GAMIL3, with an approximate 2° (180×80) horizontal resolution, serves as the atmospheric component of the Flexible Global Ocean-Atmosphere-Land System Model Grid-point Version 3 (FGOALS-g3), which participated in CMIP6 (Li et al., 2020b). For this study, the model's horizontal resolution is refined to about 1 $^{\circ}$  (360  $\times$ 160), with 26 vertical σ-layers extending to the model top at 2.19 hPa. To ensure numerical stability at the higher resolution, the dynamical core time step is reduced from 120s to 60s, while the physical parameterizations and their time step (600s) remain unchanged. As in many other climate models (e.g., Santos et al., 2021; Wan et al., 2021; Schneider et al., 2024), the performance of GAMIL3 is sensitive to the resolution, the model time step, and the coupling frequency between dynamics and physics. Therefore, it is necessary to re-tune the uncertain parameters for the new 1° configuration. During optimization, each model simulation is performed for 15 months, forced by observed sea-surface temperature (SST) and sea ice, in an Atmospheric Model Intercomparison Project (AMIP) experiment (Eyring et al., 2016). The period runs from 1 October 2010 to 31 December 2011 (hereafter referred to as AMIP2011), with the first 3 months excluded for model spin-up, leaving 12 months for analysis against observations. This method is commonly used for model uncertainty quantification and parameter tuning (Yang et al., 2013; Xie et al., 2023; 2025). After optimization, the parameter set that best fits the observations is referred to as the optimized parameter set. We use this to conduct a 10-year AMIP simulation from January 1, 2005, to December 31, 2014 (hereafter referred to as AMIP2005-2014), enabling comparison with observed climate data. To assess whether tuning atmospheric parameters results in a reasonable coupled model, the GAMIL3 atmospheric model is coupled with land (CAS-LSM; Xie et al., 2020), ocean (LICOM3; Yu et al., 2018), and sea ice (CICE4) models, consistent with the configuration used in FGOALS-g3 (Li et al., 2020b), which participated in CMIP6. A 30-year piControl simulation (Eyring et al., 2016) was then conducted to assess the model's long-term energy balance and stability under constant pre-industrial forcings. This experiment tests whether parameters performing well under observed forcings in AMIP simulations—such as prescribed SSTs, sea ice, and greenhouse gases—can also improve coupled performance. In AMIP runs, the TOA energy imbalance mainly results from greenhouse gases forcing, which traps outgoing longwave radiation. Under piControl conditions, where pre-industrial greenhouse gas concentrations are fixed, this radiative effect is absent; thus, if the AMIP-tuned parameters are physically consistent, the coupled model should yield a near-zero TOA net flux. The initial condition for the atmospheric model was the climatological mean state from atmospheric reanalysis (default configuration), while the ocean model was initialized from the equilibrated state of an OMIP simulation (a long ocean-only run forced by atmospheric reanalysis). The land model was not provided with a prescribed initial condition; instead, its state was generated dynamically during the coupled integration. To minimize the influence of potential initialization drift, the first 15 years were treated as a spin-up period and excluded from the analysis. Lastly, three additional sensitivity experiments, varying the initial values of the first 10 parameters mentioned above, are carried out to examine the impact of initial parameter selection on the optimization results. These three cases are referred to as the "random1", "random2", and "random3" cases in the captions of all relevant figures. All experiments conducted in this study are illustrated in Fig. 2

### 2.4 Covariance matrices for observations and model






























Two covariance matrices need to be prepared before the optimization process begins. The first matrix assesses the internal variability of the model system ( $C_i$ ). To derive this, perturbed initial condition experiments are conducted. In this study, these experiments involve running a total of 20 simulations, each with the three-dimensional atmospheric temperature initial state perturbed by increments of +1e-20, while all other settings remain identical to those used in the optimization. This design ensures that simulated observations within the range of internal variability receive reduced penalties,

guiding the optimization to correct systematic biases while avoiding overfitting to random climatic fluctuations. The second matrix estimates the uncertainty of observations ( $C_0$ ), which set to be diagonal, assuming no correlation between different observations, and its values are derived from absolute difference between the two available datasets for each variable after regridding and area-weighting. Specifically, data from ERA5 and National Center for Environmental Predictions/Department of Energy (DOE) 2 Reanalysis dataset (NCEP2; Kanamitsu et al., 2002) are used to derive the observation error for variable T500, RH500, and MSLP. Precipitation data from CRU and Global Precipitation Climatology Project (GPCP; Adler et al., 2003) are used for Land Precipitation (Lprecip). Data from CRU and Berkeley Earth Surface Temperature (BEST; Muller et al., 2013) are used for LAT. For the four radiation variables (OLR, OLRC, RSR, and RSRC), uncertainties are based on the estimates from Loeb et al. (2018). Both matrices contribute to the total uncertainty in the variables relative to the target observations. The total covariance matrix C is composed of the two uncertainties introduced above, calculated as:

$$C = C_0 + 2C_i \tag{1}$$

Consistent with Tett et al., (2022), we account for internal variability in both model simulations and observations by doubling the model-based estimate, reflecting a conservative assumption of comparable noise contributions. During optimization, all observation values are standardized using the square root of the diagonal elements of matrix C.

#### 2.5 Evaluation methods

The cost function F(p) is used to measure the difference between the simulated values S and the target observations O based on the parameters p. The cost function is given by:

$$F^{2}(p) = \frac{1}{N}(S-O)^{T}C^{-1}(S-O)$$
 (2),

where S is the simulated values; O is the target (observed) values; N is the number of observations;  $(S - O)^T$  is the transpose of the difference between simulated and observed values;  $C^{-1}$  is the inverse of the covariance matrix C discussed above. This

cost function quantifies how far the simulation is from the observations, considering the uncertainty (through C) and correlation between different observations. The cost function can be modified to include additional constraints, such as the net radiation flux at the TOA, along with global averages for surface air temperature and precipitation.

The Jacobian matrix, J, defined as the partial derivatives of the simulated outputs with respect to the parameters being optimized, is used to assess the influence of tuning parameters on the simulated variables. For each simulated model output  $S_i$  and parameter  $p_j$ , the Jacobian element  $J_{ij}$  is given by:

$$J_{ij} = \frac{\partial S_i(p)}{\partial p_j} \tag{3}$$

This measures how much a small change in the parameter  $p_j$  will affect the simulated model outputs  $S_i(p)$ , revealing the impact of each parameter on the variables and providing insights into their sensitivity. The Jacobians are normalized by the parameter range and internal variability. Further details about the cost function and the Jacobian are available in Tett et al. (2017).

In order to assess the extent to which the optimization has improved the performance of the simulated values, the ratios (Z) of the difference between the optimized and the default one to the standard error was adopted:

$$Z = \frac{|V_{\text{Default}} - V_{\text{Observation}}| - |V_{\text{Optimized}} - V_{\text{Observation}}|}{Standard\ error}$$
(4)

The  $V_{\rm Observation}$   $V_{\rm Default}$ , and  $V_{\rm Optimized}$  represent the observation value, simulated values using the default and optimized parameter sets, respectively. The *Standard error* represents the observation error of the corresponding variables. Improvement is expected for the variable if Z > 0, while if Z < 0, no improvement is anticipated, and performance may even worsen.

## 2.6 Optimization algorithm

The challenge of optimizing the model parameters numerically lies in the high computational cost and potential noise associated with model evaluations, making traditional derivative-based optimization methods impractical. There are several optimization algorithms the system provides, such as (derivative-free) Gauss-Newton variants, the pySOT algorithm, and the DFO-LS algorithm. We use the DFO-LS

algorithm as it appears to have better performance in model calibration (Oliver et al., 2022, 2024; Tett et al., 2022) relative to other algorithms such as Gauss-Newton (Tett et at., 2017) or CMA-ES (Hansen, 2016). This algorithm is a sophisticated optimization method designed to handle nonlinear least-squares problems without requiring derivative information. This algorithm is particularly useful in scenarios where function evaluations are expensive or noisy. Inspired by the Gauss-Newton method, DFO-LS constructs simplified linear regression models for the residuals, allowing it to make progress with a minimal number of objective evaluations (Cartis et al., 2019).

The underlying algorithmic methodology for the DFO-LS algorithm is detailed in Cartis et al. (2019). Here, we provide a brief overview of the algorithm, with a detailed description of its parameter settings available in Supplementary S1. The optimization problem is defined as minimizing the sum of the squared residuals

$$f(p) := \frac{\sum_{i=1}^{N} r_i(p)^2}{N}$$
 (5),

where r(p) represents the differences between model outputs and observations; in our case,  $r_i(p) \coloneqq C^{\frac{1}{2}}(S_i - O_i)$ . DFO-LS approximates the residuals without derivatives by creating a linear regression model at the current iteration. DFO-LS employs a trust region framework for stable optimization, which dynamically adjusts the search region to balance exploration and exploitation. After constructing the regression model, the algorithm solves the trust region subproblem to determine the step size and direction for updating parameters. The actual versus predicted reduction in the cost function is calculated to decide whether to accept or reject the step, with adjustments made to the trust region size accordingly. The algorithm follows these steps: initialization of parameters and trust region, model construction at each iteration, solving the trust region subproblem, accepting or rejecting steps, updating the interpolation set, and checking termination criteria. This structured approach ensures robust and efficient optimization in minimizing model discrepancies.

### 3 Results

#### 3.1 AMIP2011 simulations

## 3.1.1 GAMIL3 10-parameter case

The first experiment aims to optimize the ten sensitive parameters related to convection and microphysics parameterization schemes (Table 2). In this experiment, several parameters—such as *ke* and *captlmt*—changed significantly from their default values, while *cmftau* and *c0* showed only small changes (Fig. 3a). Fig. 3b shows the progression of the cost function over iterations for the 10- and 20-parameter cases. Note that the cost function is divided by the number of observations, and a smaller cost function indicates better simulation accuracy against observations. In the 10-parameter case, the optimization required 29 total model evaluations (11 initial perturbation runs + 18 iteration runs), reaching the lowest cost function value of approximately 3.5. The cost function drops rapidly from about 7.5 to 3.5 in the first iteration run, followed by a slower decline with some fluctuations.

Fig. 4 shows the reduction or increase in simulation error in terms of the number of standard errors through optimization. In the 10-parameter case (solid dots), 24 out of 34 variables (approximately 71%) show Z values greater than zero, indicating improved performance against the default case. Moreover, for 11 of these 24 variables, the optimization reduced the error by more than 1 standard error, with 5 of these showing improvements greater than 3. This is particularly evident in the RSR, MSLP, and the tropical variables of T500. While most variables can be effectively tuned, several variables, such as OLR, OLRC, and LAT, are worse than the default case. However, except for LAT\_NHX, the performance of these variables did not degrade by more than one standard error. The blue dots in Fig. 5 represent the global area-weighted mean of different variables for the tuning year (2011) in the 10-parameter case. Comparing to the observational values, the optimization successfully improved most variables (9 out of 10), bringing them closer to the observations. Although some variables showed slight deviations from the observations after optimization, nearly all remained within their uncertainty range (except for OLRC), which is also reasonable in model tuning.

Since the cost function is a simple statistical indicator of the distance between the area-weighted mean of the simulations and the observations, analyzing the spatial

distribution of the variables is crucial when evaluating the performance of the optimized parameter sets. Fig. 6a presents Taylor diagrams for all tuning variables under three parameter cases for the optimized year (2011). The results indicate that, compared to the default case (yellow), most variables' performance improved to varying degrees in the 10-parameter case (blue). For instance, while the standard deviation (SD) of the MSLP in the default result was much closer to the observations, the 10-parameter case exhibited a larger pattern correlation (PC) coefficient and a smaller root mean square deviation (RMSD). Some variables, including Lprecip, NETFLUX, and T500, showed improvements in all three metrics (SD, PC, and RMSD). However, other variables, such as OLR and RH500, showed slight deterioration after optimization, as partially suggested in Fig. 4.

The "optimized" parameter set referred to in this study is the set where the cost function reaches its lowest value. However, the robustness of this parameter set, compared to others with similar cost function values, remains to be evaluated. To address this, two additional experiments were conducted (Table S1 and Fig. S1), selecting parameter sets with cost function values closest to the optimized one to evaluate the potential impact of this choice. Table S1 shows that the parameter values for the two sets (Experiment1 and Experiment2), which have cost function values close to the minimum (Optimized), are quite similar, particularly for Experiment1, which has the closest cost function value. The results from the AMIP2005-2014 simulations show that, while most variables exhibit similar behaviors to those of the Optimized set, notable differences are observed in T2M and Lprecip. Overall, although differences in model behavior arise from the choice of the optimized parameter set, these differences are not substantial enough to significantly alter the model's performance.

## 3.1.2 GAMIL3 20-parameter case

To investigate the impact of different numbers of tuning parameters on optimization and the robustness of the tuning results, additional 10 parameters related to microphysics and turbulence schemes (Table 2) were included alongside the existing 10 parameters. In the 20-parameter case, the initial perturbations for the original 10

parameters were kept the same as in the 10-parameter case to ensure a fair comparison. Comparing the optimal values of the 20-parameter case with the default values shows that several parameters had large changes. Parameters such as c0 conv, ke, capelmt, dzmin, Dcs, and ecr showed significant deviations from their default values (Fig. 3a). Comparing the two sets of optimal parameters reveals both differences and consistencies. While most parameters, such as *capelmt*, *alfa*, and *rhcrit*, change in the same direction and display similar magnitudes, some parameters, like ke and cmftau, are adjusted in the opposite direction. These differences may be attributed to the compensating errors within in the model, where adjustments to one parameter can offset or amplify the effects of another—a phenomenon further explored in Section 3.3. When examining the tuning procedure (Fig. 3b), it is evident that the cost function dropped rapidly to a value very close to the minimum in the first iteration run, similar to the 10parameter case. The system required a total of 31 runs (21 initial perturbation runs + 10 iteration runs) to reach the lowest cost function value (2.87), which is only two more than that required for the 10-parameter case. This suggests that adding ten additional parameters increases the total number of evaluations only marginally, indicating that when optimizing with DFOLS, there is no need to be overly selective about parameter choice. The minimum cost achieved is comparable to that of the 10-parameter case, with fewer additional runs required after the initial perturbation phase to reach the minimum. This implies that including more tuning parameters has a small impact on the total cost but enhances tuning efficiency. This improvement can be attributed to the inclusion of additional parameters related to other parameterization schemes, which enhances model tuning and yields more realistic results compared to observations.































Comparing the Z values from the 20-parameter case to those from the 10-parameter case (Fig. 4), we find that 25 out of 34 variables (approximately 74%) have Z values greater than zero, slightly higher than in the 10-parameter case. Among these, 11 variables show improvements of more than 1 standard error, with 6 exhibiting significant improvements of over 3 standard errors (notably in T500 and MSLP), which is also better than the 10-parameter case. While most variables in the 20-parameter case demonstrate equal or greater improvements than in the 10-parameter case, some, like

OLR and OLRC, perform worse. The global area-weighted mean of all variables (shown by red dots in Fig. 5) indicates that, except for OLR, RH500 and Lprecip, variables improved compared to the default case. Although RH500 shows a greater deviation from observation, it still falls within the uncertainty range. Significant differences between the 20-parameter and 10-parameter cases are observed in the two radiation variables (OLR and RSR) and the two surface-related variables (T2M and Lprecip). These differences may partly result from certain parameters compensating for each other, which will be discussed later. The Taylor diagram in Fig. 6a shows that most variables have improved compared to the default case. Relative to the 10-parameter case, OLR, RSR, RSRC, MSLP, and Lprecip perform better in the 20-parameter case. However, NETFLUX and T2M perform worse.

#### **3.2 AMIP2005-2014 simulations**






























Although our cost function explicitly accounts for internal variability (Eq. 1), tuning and evaluating the model using only a one-year simulation may still introduce uncertainties due to atmospheric internal variability (Bonnet et al., 2025), such as phase shifts in the North Atlantic Oscillation (NAO) or stochastic tropical convection patterns like the Madden-Julian Oscillation. Therefore, a longer simulation with adjusted parameter settings using AMIP drivers is necessary to assess the robustness of the tuning across different phases of intrinsic variability. Thus 10-year simulations from 1 January 2005 to 31 December 2014 are conducted for the default and two optimized parameter sets. Compared to the results from 2011, the average AMIP2005-2014 results (Fig. 4b) show no significant differences between the two cases, as both exhibit similar changes across most variables. For example, T500 and RSR show much improvement in both cases, while OLR and OLRC perform worse. However, several variables show For differences between the conditions. instance. while the two MSLP TROPICSOCEAN DGM shows an improvement of more than 20 standard errors relative to observations in the 2011 simulation with the 10-parameter case, it deviates from the observation by over 10 standard errors in the 10-year simulation. Additionally, while the 20-parameter case demonstrated improvement in the 2011

simulation, its performance declined in the 10-year simulation. This temporal inconsistency suggests that certain parameter adjustments may be sensitive to the specific climate state of 2011, which was characterized by a moderate La Niña. In contrast, variables such as T500, RSR, and NETFLUX exhibit consistent improvements across both simulations, indicating a robust response to parameter tuning that is less dependent on interannual variability.































The time series of the AMIP2005-2014 simulations in Fig. 5 show that, for the 10parameter case, 8 out of 10 variables are either much closer to the observations or very similar (OLR, OLRC, and RSRC) to those in the default case. Only two variables, RH500 and Lprecip, are slightly further from the observations but still within uncertainty. The most striking finding is the improvement of the variables related to the energy balance of the climate system (RSR and NETFLUX). For the default case, due to the large outgoing shortwave radiation, NETFLUX has an error of about 5 W/m<sup>2</sup>. In addition, T500 in the default case is too cold by almost 2K. After optimization, while OLR shows little change, RSR decreased by nearly 5 W/m<sup>2</sup>, considerably reducing the model bias and leading to smaller biases in NETFLUX and T500. Furthermore, the results suggest that MSLP, RSRC and OLRC are hard to tune. In the 20-parameter case, compared to the default, all variables-except RH500, OLR, T2M and Lprecip-show either reduced biases or biases that are very close (OLRC and RSRC) to those in the default case. Both OLR and Lprecip perform notably worse than in the default case, with both variables being too low compared to the observations. This is less successful, in relative terms, than the 10 parameter case, where 8 variables exhibit reduced or similar bias relative to the default. However, T500 and the MSLP-two variables that deviated significantly from the observations in the default and 10-parameter cases have been further tuned and now align more closely with observation.

Similar to the Taylor diagram of the AMIP2011 results, the AMIP2005-2014 simulations (Fig. 6b) also demonstrate varying degrees of improvement across the three metrics for most variables in both optimized cases. For instance, both cases improve all three metrics for Lprecip, NETFLUX, and RSRC compared to the default case, consistent with the AMIP2011 results. While Lprecip, RSRC, T2M, and NETFLUX in

both optimized cases exhibit similar behavior to the AMIP2011 results, MSLP, RH500, and RSR behave differently. Comparing this with Figs. 4 and 5, the results suggest that this tuning yields only minor improvements to the spatial patterns of the variables but primarily reduces their biases relative to observations. Examining zonal averages (Fig. 7) reveals more specific details, particularly the differences between tropical and extratropical regions. T500 and RSR have large tropical biases which tuning considerably reduces. In contrast, RH500, OLR, RSRC, and MSLP have larger biases in extratropical, especially polar regions. These regional biases may come from uncertainties in complex high-latitude processes, such as sea ice and snow cover feedback mechanisms, which are not well represented in the model (Goosse et al., 2018). Across the three cases, average performance is similar to that found earlier, with T500, RH500, OLR, RSR, T2M, and Lprecip most affected by tuning and most sensitive to parameter changes, while OLRC, RSRC, and MSLP are little impacted by optimizing. Specifically, MSLP is highly sensitive to unresolved gravity wave drag processes (Sandu et al., 2015; Williams et al., 2020), which were not included in our parameter tuning. Previous experiments with the IFS model indicate that increasing orographic and surface drag in the Northern Hemisphere can reduce MSLP biases (Kanehama et al., 2022). While the global mean OLRC is similar across cases due to regional compensation (Fig. 5d), the meridional distribution reveals notable differences (Fig. 7d). In the tropics, increased upper tropospheric water vapor—particularly in the 20-parameter case (Fig. 9a-9b) enhances the greenhouse effect and reduces outgoing clear sky longwave radiation. In contrast, decreased water vapor in high-latitude regions, especially in the 20-parameter case, leads to increased OLRC. RSRC remains nearly unchanged across all simulations due to the use of identical surface albedo. Additionally, while changing physical parameters generally affects the entire atmosphere, some variables respond differently in specific regions. For example, RH500 shows a more pronounced response in tropical regions, while land T2M responds more noticeably in the extra-tropics.

# 3.3 Impacts of tuning on GAMIL3






























What parameters and processes would affect these model tuning behaviors? As

shown in Fig. 8, parameters such as c0 conv, cmftau, rhcrit, rhminl, rhminh, and Dcs affect significantly simulated variables, particularly NETFLUX, Lprecip TROPICSLAND, RSR TROPICSOCEAN, OLR TROPICSOCEAN, and TEMP@500. Notably, most of these parameters have also been adjusted significantly in the 10- and 20-parameter cases compared to the default. rhcrit defines the RH threshold for triggering deep convection and is a parameter with a strong influence on RH. Fig. 3a shows that *rhcrit* decreased from the default case, whose value is 0.85, to the 10-paramter case and 20-parameter case, whose values are 0.83 and 0.82, respectively. A lower *rhcrit* significantly promotes deep convection by reducing the triggering threshold, which enhances water vapor transport from the lower to the mid and upper atmospheric layers. This could lead to a drop in RH below troposphere and a rise above it (Fig. 9a). This effect is especially pronounced in the tropics, where deep convection dominates vertical moisture transport (Fig. 5b, 7b, and 9b). While a lower rhcrit threshold would theoretically enhance precipitation by promoting deeper convection, our simulations instead show an overall decrease in precipitation. This apparent discrepancy suggests the parameter's effect is modulated by compensating atmospheric processes. Specifically, enhanced vertical moisture transport (Fig. 9a-9b) reduces low-level humidity availability, thereby weakening updrafts and ultimately decreasing total precipitation (blue line in Fig. 5h).































A deficit in low-level cloud fraction is evident in Fig. 9c-9d, primary due to the increase in *rhminl* from the default value of 0.95 to 0.97 and 0.96 in the 10- and 20-parameter cases, respectively. Although the 10-parameter case has a higher threshold for low level cloud formation than the 20-parameter case, Fig. 9c-9d shows the different result, which can be explained by the compensatory effects of other parameters. Optimized results indicate that *cmftau*, another key parameter, has a lower value in the 20-parameter case (~4284) compared to the default (~4800) and the 10-parameter case (~4931). This decrease in *cmftau* likely strengthens shallow convection while weakening deep convection, reducing upward water transport and RH throughout the troposphere, contributing to the decreased low-level cloud fraction (Xie et al., 2018) and further reducing precipitation (Fig. 5h). Consequently, the lower low-level cloud

fraction in the 20-parameter case, compared to the 10-parameter case, reflects the compensatory effects of these key parameters, with the influence of the reduced *cmftau* outweighing that of *rhminl*. Low-level clouds strongly reflect shortwave radiation, producing a cooling effect. Therefore, a reduction in low-level clouds allows more shortwave radiation to penetrate the lower atmosphere, reducing outgoing shortwave radiation to space (blue lines in Fig. 5e and 7e) and warming the region (blue lines in Fig. 5a and 7a; Fig. 9e), including near the surface (blue lines in Fig. 5g).































Comparing the 20-parameter case to the default case, the tuning results show that one sensitive parameter, Dcs—the autoconversion size threshold for ice to snow—has been significantly increased. This adjustment suggests that a higher Dcs leads to increased RSR and T2M, while also resulting in lower OLR and Lprecip (Fig. 8). ccrit, which sets the minimum turbulent threshold for triggering shallow convection, affects both OLR and Lprecip in a manner similar to Dcs. Specifically, clouds with higher ice content trap more OLR from the Earth's surface, potentially amplifying the greenhouse effect by retaining more infrared radiation (red lines in Fig. 6c and 8c). This results in a warming effect, particularly at lower atmospheric levels and even near the surface, especially during nighttime or in polar regions (red lines in Fig. 5a, 5g, 7a, and 7g; Fig. 9f). Additionally, raising the autoconversion threshold from ice to snow is expected to allow more ice to remain in the atmosphere, directly leading to a reduction in precipitation (red line in Fig. 5h), and increased cloud optical thickness, thereby enhancing the reflection of incoming shortwave radiation. This enhanced reflectivity partially offsets the impact of reduced low-level cloud cover on the RSR in the 20parameter case, leading to a smaller decrease in RSR compared to the 10-parameter case (Fig. 5e and 7e), consistent with known radiative differences among cloud types (Chen et al., 2000). Increasing *ccrit* suppresses shallow convection by requiring stronger turbulence to initiate cloud formation, thereby reducing low-level cloud cover. This reduction enhances outgoing longwave radiation and surface solar heating, which in turn promotes evaporation and increases Lprecip. Therefore, adjusting Dcs and ccrit in future work may offer a promising approach for improving the simulation of OLR and Lprecip, both of which are underestimated relative to the default case.

## 3.4 Coupled model evaluation



In order to evaluate the performance of different parameter sets in long-term 649 climate simulations, it is essential to apply them to a coupled model. To assess the 650 impacts of atmospheric parameter tuning on coupled model performance, we conducted 651 a 30-year piControl simulation using GAMIL3 coupled to land, ocean, and sea ice 652 components (see Methods 2.2), analyzing the final 15-year period after model spin-up. 653 In the default case the model starts with a large negative NETFLUX of around -4 654 W/m<sup>2</sup> (Fig. 10a), consistent with the results in Fig. 5j, indicating that the climate system 655 is losing energy at this stage. As the model integrates, the NETFLUX increases, 656 approaching zero after approximately five model years, achieving a stable energy 657 budget for the remaining simulation period. This change in NETFLUX is found to be 658 almost equally driven by a ~2 W/m<sup>2</sup> reduction in both RSR (Fig. 10b) and OLR (Fig. 659 10c) simultaneously. However, despite these radiation variables, particularly the 660 NETFLUX, approaching a stable state, the ocean continues to lose energy rapidly (Fig. 661 10d) with no signs of stabilization by the end of the simulation. For T2M (Fig. 10e), 662 the simulated values in the piControl run deviate significantly from the target range of 663  $13.6 \pm 0.5$ °C (Williamson et al., 2013). While the decrease in OLR is physically 664 consistent with the cooling of T2M, the reduction in RSR is primarily attributed to 665 oceanic adjustment processes. In particular, a cold SST bias (Fig. S3b) induced by the 666 original parameter settings leads to a rapid decline in low-level cloud cover over 667 tropical and subtropical ocean basins—especially in the western Pacific warm pool 668 region and the South Atlantic (Fig. S3c). Most areas of cloud reduction spatially 669 coincide with regions of diminished reflected shortwave radiation (Fig. S3d), a 670 relationship further supported by changes in shortwave cloud forcing (SWCF; Fig. S3e). 671 Overall, although the NETFLUX appears to reach a stable state, the system continues 672 to lose energy and remains far from the tuning target in the default case. 673 For both optimized cases, the NETFLUX (Fig. 10a) remains stable throughout the 674 30-year simulations, with values of about 2 W/m<sup>2</sup>. Although not exactly reaching the 675

target of 0 W/m<sup>2</sup>, they are still within the model spread range of -3 to 4 W/m<sup>2</sup> (Mauritsen

et al., 2012). Further analysis revealed that the relatively large energy imbalance primarily originates from the GAMIL3 atmospheric model, which exhibits a persistent imbalance of approximately 1.4 W/m<sup>2</sup> in its AMIP configuration—a feature also observed in the piControl runs—due to non-conservation in the dynamical core. This systematic issue is consistent with other atmospheric or coupled models (e.g., up to 1.0 W/m<sup>2</sup> for CAM6 at 1° resolution (Lauritzen and Williamson, 2019), 1.3 W/m<sup>2</sup> for FGOALS-g3, and 3.3 W/m<sup>2</sup> for INM-CM4-8, calculated from Wild, 2020). Notably, this energy leakage nearly identical (±0.1 W/m<sup>2</sup>) between the default and optimized runs, indicating that the model improvements, such as reduced climate drift, result from genuine parameter tuning rather than compensation for the energy bias. This conclusion is further supported by the coupled model's stabilized energy budget following the spinup period (Fig. 10). The change in NETFLUX in the 10-parameter case is primarily driven by a decrease in RSR (Fig. 10b), while in the 20-parameter case, it is mostly due to a reduction in OLR (Fig. 10c), consistent with the results in Fig. 5c and 5e. Both the volume-averaged ocean temperature (Fig. 10d) and the T2M (Fig. 10e) exhibit a slight initial adjustment during the initial few years, followed by stabilization. Drift may occur during the initial integration period due to inconsistencies between the OMIP-forced ocean state and the reanalysis-based atmospheric initial conditions. However, in both cases using atmosphere-optimized parameters, the system stabilized rapidly, and neither the TOA net flux nor ocean temperature exhibits significant trends beyond the initial adjustment period of a few years. A small long-term drift is still evident in Fig. 10d, which may be related to the adjustment of deep ocean processes. This demonstrates that the parameters optimized for the atmospheric model remain effective in the coupled system configuration, with no clear evidence of compensation for ocean-related drift.































Results from the simulated SST biases in Fig. 11a–11c for the default case show strong cold biases relative to observations, with maximum deviations exceeding -4°C over the North of Pacific and Atlantic. The simulated SST biases in Fig. 11d–11i indicate that both optimized cases show substantial improvement over the default case in terms of SST patterns and deviations, although some negative deviations in the northern Pacific and Atlantic persist—a common issue for most GCMs (Zhang and

Zhao, 2015a; Wang et al., 2018). Previous findings suggest that the two optimized cases exhibit cloud fraction significantly different from the default case, with simulated radiation improvements primarily observed in shortwave radiation for the 10-parameter case and in longwave radiation for the 20-parameter case. Therefore, it is necessary to investigate the shortwave and longwave cloud forcing in these two cases (Fig. 12). The results for both cases show that the combined effect of these two cloud forcings acts as a significant positive influence globally, contributing to the flux of energy towards the ocean and increasing ocean temperature. Specifically, the shortwave cloud forcing has a greater weight than the longwave in the 10-parameter case, mainly due to the parameters *rhcrit* and *rhminl*, as mentioned earlier. In contrast, the longwave cloud forcing outweighs the shortwave in the 20-parameter case, primarily due to the effects of *Dcs*. While the shortwave cloud forcing exerts a negative effect over the tropical ocean, the longwave cloud forcing provides a significant compensatory effect. A similar behavior is observed in the 20-parameter case.

Overall, the two optimized cases result in a more realistic coupled model, not only maintaining the model's energy balance and reducing climate drift, but also improving the simulated ocean state, such as SST distribution. Although the two optimized cases exhibit different behaviors—with the 10-parameter case showing lower RSR and the 20-parameter case showing lower OLR—tuning has allowed them to achieve stability through distinct mechanisms. While we acknowledge that multi-century integrations would provide additional insight into the model's equilibrium climate response, our primary goal was to test whether AMIP-tuned parameters remain valid in a coupled setup. For this purpose, a 30-year piControl run is scientifically adequate. The results show that the model quickly reaches energy balance stability for both the 10- and 20parameter cases (TOA net flux drift < 0.05 W m<sup>-2</sup> per decade) and that ocean heat content drift remains minimal (< 0.008 ° C per decade) after year 15, indicating that the system achieves a quasi-equilibrium state. This timescale is reasonable, since the upper ocean—where much of the adjustment occurs—has a relatively short adjustment timescale of about 1-5 years. The stabilized climate indicators and consistent system behavior (Figs. 9 and 10) confirm that the tuned parameters yield a credible coupled climate without introducing systematic drifts. Similar integration lengths have been used in other studies (e.g., Tett et al., 2017). While longer runs could refine the equilibrium further, they are unlikely to change our main conclusion that the parameter transfer is robust.

# 3.5 Sensitivity of initial parameters

As stated in the previous section, the initial parameter values used for tuning are primarily informed by expert judgment, which has been recognized as crucial and necessary in other studies (Hourdin et al., 2017; Williamson et al., 2017; Jebeile et al., 2023; Lguensat et al., 2023). To further investigate the extent to which initial parameter choices influence tuning results, we conducted three additional sensitivity experiments with randomly selected initial parameter values (Table S2), focusing on the first 10 parameters.

The optimized parameter values in these randomized experiments (represented by stars in Fig. 3a) exhibit significantly larger spreads compared to the default and original optimized values (blue dots), particularly for parameters such as c0\_conv, capelmt, and c0, which nearly span their entire plausible ranges. This finding indicates that the model could reach entirely different optimized states depending on initial values. During the tuning process, the cost function (Fig. 3c) for these cases exhibited a rapid decrease, stabilizing at similar values across all three experiments after approximately 10 iterations, with an additional 10–20 runs required to reach the optimized state. This pattern further demonstrates the efficiency and robustness of the tuning algorithm.

Given the substantial differences in the optimized parameters, it is worthwhile to further investigate their Jacobian differences to gain a more comprehensive understanding of each parameter's impact on the variables. Fig. 13 shows the Jacobian ranges for four cases (including the original optimized case), with Jacobian calculated around the optimized parameter set for each case. The results generally demonstrate consistency with the parameter sensitivities shown in Fig. 8. Variables sensitive to most parameters exhibit substantial variability, while highly sensitive parameters, such as  $c0\_conv$ , cmftau, rhcrit, rhminl, and rhminh, introduce considerable uncertainty across

multiple variables, depending on their initial values and interactions with other parameters. Conversely, RSRC and OLRC remain largely insensitive to parameter changes, whereas MSLP, NETFLUX, Lprecip, and TEM@500hPa are influenced by most parameters, also aligning with the findings in Fig. 8.

The performance of these three optimized parameter sets in the AMIP2005-2014 simulations is shown in Fig. S2. Generally, NETFLUX was most closely aligned with observations across all cases, primarily due to the additional constraint incorporated into the tuning algorithm. However, notable differences across different cases remain, with each case following a distinct optimization pathway, though most results still fall within uncertainty ranges. For example, the third experiment achieved the closest alignment for T500 but at the expense of T2M and Lprecip compared to other cases, highlighting inherent trade-offs and model structural errors that hinder simultaneous optimization of these variables. As seen in prior findings, RSRC and MSLP proved difficult to tune, while OLRC was adjustable but deviated in the opposite direction from observations, accompanied by a discrepancy in RH500 alignment.

Overall, these sensitivity experiments confirm the efficiency of the tuning algorithm and underscore the importance of expert judgment in selecting initial parameter values. Expert selection not only ensures satisfactory model performance at the start of tuning but also enhances tuning effectiveness, even though structural errors in the model remain.

#### 4 Discussion

In this study, we developed an objective and automatic parameter tuning framework using the Derivative-Free Optimizer for Least-Squares (DFO-LS) method to tune the newest version of the Grid-Point Atmospheric Model (GAMIL3). The results highlight the effectiveness of this method in tuning atmospheric parameters, particularly those initially set based on expert judgment, as demonstrated by notable improvements in model accuracy across multiple variables and enhanced climate system stability. However, several aspects of this work require further clarification.

Firstly, as noted earlier, the 'optimized' parameter set in this study refers to the set

at which the cost function achieves its minimum value. However, results in Figs. 3b and 3c indicate that, for each case, there are several cost function values close to this minimum. We have shown that these differences are not substantial enough to significantly alter the model's performance. However, this finding suggests that parameter ranges associated with similar cost function values may provide valuable insights into the acceptable parameter space for model optimization. We acknowledge that focusing exclusively on minimizing cost function values to obtain a single optimized parameter set during tuning can increase the risk of overfitting and compensating errors, which is a common challenge in model tuning. Although the results of this study show no clear signs of overfitting—both the 10- and 20-parameter optimized cases, starting from expert-judged initial values, ultimately produce reasonable coupled model results—it remains important to carefully consider potential overfitting impacts.

Secondly, this study shows that tuning either different numbers of parameters or varying initial parameter values can yield diverse optimized results, each improving certain aspects of the model. This suggests that although tuning can lower the cost function to comparable levels, the final tuned state of the model is not necessarily unique—a common issue encountered in model tuning (Hakkarainen et al., 2013; Hourdin et al., 2017; Eidhammer et al., 2024), likely due to the compensating errors within the model and uncertainties in the observational data. On one hand, introducing constraints, such as assigning greater weight in key variables during tuning, could help achieve more realistic results. For instance, applying constraints on NETFLUX during tuning ensures consistently good performance across all the cases in the AMIP2005-2014 simulations. In the 20-parameter case, adding constraints on OLR and RSR would maintain their performance while also improving T500 and MSLP. On the other hand, while different parameter sets satisfied the lowest cost function in different ways, it is important to remember that the cost function is simply a statistical measure of the distance between the area-weighted mean of the simulations and observations. Therefore, a comprehensive evaluation is essential to identify the most suitable parameter set (Eidhammer et al., 2024). Beyond minimizing cost function values and aligning statistical indicators with observations, it is crucial to evaluate the spatial distributions of variables, the equilibrium state of the climate system in coupled models, and the model's climate sensitivity (Tett et al., 2022; Eidhammer et al., 2024). These aspects should be further evaluated to ensure robust model performance.































Thirdly, while our 1-year optimization produced parameters that remain effective in extended runs (as shown by the AMIP2005–2014 and 30-year piControl validations) and internal variability was explicitly accounted for in the cost function (Eq. 1), including interannual variability—using a longer tuning period like the 5-year approach of Tett et al. (2022)—could further improve results, especially for variables with large interannual variability (e.g., MSLP, Lprecip) and dynamical outputs sensitive to the chosen year. This is supported by Bonnet et al. (2025), who show that short-term tuning works well for physical variables with low interannual variability but multi-year tuning better captures dynamical variability. Based on Bonnet et al. (2025) and our own results—such as the difference observed between 1-year and 10-year simulations for MSLP\_TROPICSOCEAN\_DGM, which degraded from +20 $\sigma$  to -10 $\sigma$ —we might expect approximately 10–20 % better performance for variables that are particularly sensitive to interannual variability, such as tropical precipitation patterns or extratropical circulation indices, since a longer tuning period would better sample different climate regimes and reduce sensitivity to single-year anomalies. However, longer tuning greatly increases computational cost—about 5–6 times higher for 5-year runs. Our current strategy balances efficiency and robustness, but certain metrics like T2M and Lprecip might still benefit from longer tuning. This trade-off warrants further study, particularly where an accurate representation of interannual variability is crucial. Lastly, to assess how the number of tuning parameters affects the optimization

Lastly, to assess how the number of tuning parameters affects the optimization process, we used the same initial perturbation runs for the ten shared parameters in both the 10- and 20-parameter cases, enabling a consistent evaluation of their sensitivity to the simulated results. While this approach allows a straight forward comparison, it may also constrain the optimization in the 20-parameter case by introducing bias into the initial search space. To address this potential limitation, we conducted additional experiments in which all twenty parameters were initialized with independent

perturbations (Fig. S4–S6) by adjusting the *rhobeg* parameter in the DFO-LS algorithm from its default value of 0.18 to 0.23. These additional experiments yielded several important insights that strengthen our original conclusions. First, although the optimized parameter values in the new 20-parameter case differ somewhat from those in the original setup, most shift in the same direction relative to the default values (Fig. S4). Moreover, the optimization consistently converged to similar cost function values (2.68 vs. 2.87), despite differences in the initial perturbations and optimization pathways, highlighting the robustness of our tuning framework. Second, both approaches produced nearly identical simulation performance in the 10-year AMIP and 30-year piControl experiments (Fig. S5–S6), despite relying on different parameter sets. This suggests that the performance in the 20-parameter case may be dominated by a subset of the most sensitive parameters, such as *Dcs*, *rhcrit*, *c0\_conv*, and *cmftau*, which have been shown to strongly influence the simulated results. These findings provide strong evidence that our conclusions regarding the robustness of the optimization and the effect of increasing the number of tuning parameters remain valid.

Some limitations remain. For instance, although the coupled model simulations show improvements in energy stability and reduced climate drift, certain regional biases in SST persist. These biases suggest that while tuning enhances model performance, there may be systematic issues within the model's physics that cannot be fully addressed through parameter tuning alone. Resolving these regional discrepancies may require further refinement of model physics or additional modifications to the tuning framework. Additionally, the optimized cases show a relatively large TOA energy imbalance (~2.0 W/m²) despite a well-tuned NETFLUX in AMIP runs, which originates from energy non-conservation in the atmospheric model's dynamical core. In the AMIP configuration, prescribed SSTs act as an infinite energy source/sink, masking this internal leakage in the dynamical processes. By contrast, the coupled system exposes the dynamical core's non-conservation as a stable but imbalanced energy state. This interpretation is supported by our ongoing experiments (not shown) following Williamson et al. (2015b), where correcting energy conservation in the dynamical core reduced the TOA imbalance in the piControl runs to about 0.5 W m⁻² within the same

tuning framework. These results underscore that while parameter tuning can improve model fidelity, structural errors in the dynamical core—particularly its energy non-conservation—must be addressed to achieve physically consistent climate simulations. Finally, because variables such as lower tropospheric temperature, humidity, cloud fraction, and cloud radiative effects are highly sensitive to the model time step and the coupling frequency between dynamics and physics, it would be valuable to explore the tuning performance under different time step settings in future work.

#### 5 Conclusions






























The study focuses on optimizing an atmospheric model by simultaneously perturbing and tuning multiple parameters associated with convection, microphysics, turbulence, and other physical schemes. Two primary experiments were conducted using AMIP2011 simulations (2011, with 3-month spin-up): one adjusted 10 parameters and another adjusted 20 parameters. Validation was then performed through extended independent decadal AMIP (AMIP2005-2014) simulations and 30-year coupled piControl simulations. Consistent performance across timescales and model configurations confirmed that the tuning corrected systematic biases rather than overfitting. In the 10-parameter tuning, significant changes were made to several sensitive parameters, resulting in a notable reduction in the cost function and improved model accuracy. Out of 34 variables, 24 showed improved performance, although some remained challenging to optimize due to structure errors in the model. In the 20parameter tuning, additional parameters related to microphysics and turbulence were introduced, resulting in slight performance improvements for 25 out 34 variables. However, certain variables experienced a decline in performance. While the 20parameter case achieved a lower cost function more quickly than the 10-parameter case, the increased complexity required careful management of parameter interactions and compensatory effects.

To evaluate the robustness of the tuning results, we conducted AMIP2005-2014 simulations. The findings showed that the optimized parameter sets maintained their performance improvements over extended simulation periods, though variables like

MSLP exhibited variability depending on the specific period analyzed. Time series analyses indicated that the optimized models more accurately captured the energy balance of the climate system, particularly by improving the balance of outgoing shortwave and longwave radiation and stabilizing surface temperatures. However, some variables remained challenging to optimize consistently across different regions and timescales. The optimized parameter sets were further tested in a coupled model setup that integrated land, ocean, and sea ice components. The results demonstrated improved energy budget stability, reducing climate drift and leading to more realistic SST simulations. Both the 10- and 20-parameter optimizations yielded more reasonable behavior in the coupled model, though persistent regional biases, particularly in the northern Pacific and Atlantic, remained.

Three additional experiments, in which the initial values of the first 10 parameters were randomly selected, were conducted to evaluate its impact on the optimized results. The results further confirm the efficiency and robustness of the algorithm, as it rapidly minimizes the cost function after the first 10 runs, although the optimized parameter values and their performance across different cases show significant variation. Overall, these findings emphasize the importance of expert judgment in parameter selection and its role in enhancing model performance.

In conclusion, the proposed DFO-LS-based tuning framework presents a robust and efficient approach for enhancing climate model performance. By combining Jacobian estimation with sensitivity analysis, the framework quantitatively maps how parameters affect key variables and thereby exposes compensating errors between physical schemes (for example, interactions between deep convection and microphysics). These parameter–variable mappings yield direct insight into model structural uncertainties and supply objective diagnostics that guide development. When model physics are changed, the framework supports rapid retuning and systematic interversion comparison: systematic shifts in optimal parameter values then serve as concrete evidence of how structural modifications alter model behaviour. Implemented and exercised primarily by a single researcher within 12 months, the approach also demonstrates high human-resource efficiency and practical scalability. Although no

- single parameter set is expected to transfer unchanged across model generations,
- automating the exploration process transforms development from manual trial-and-
- error into an efficient, reproducible, and more objective workflow. Applied across
- GCMs, this methodology can accelerate model development, reduce parametric
- uncertainty, and improve the reliability of climate projections.
- Code availability. All codes are available on Zenodo under the DOI
- 10.5281/zenodo.14772250, with the citation: "Tett, S. (2025). ModelOptimization.
- Zenodo. https://doi.org/10.5281/zenodo.14772250"
- Author contributions. SFBT conceptualized the study, with WJD sponsoring the
- computational and necessary resources. SFBT and CC developed the optimization
- algorithm, while SFBT and WJL implemented the optimization framework under the
- guidance of LJL and CC. WJL conducted all model experiments and performed the
- necessary analyses. WJL drafted the manuscript, with all co-authors providing feedback
- and suggestions.
- *Competing interests.* The authors declare that they have no conflict of interest.
- Acknowledgements. WJL is funded by the National Natural Science Foundation of
- China (U21A6001, 42175173), the GuangDong Basic and Applied Basic Research
- Foundation (2023A1515240036), the Innovation Group Project of Southern Marine
- Science and Engineering Guangdong Laboratory (Zhuhai) (No. SML2022006), and the
- Open-end Funds of Ministry of Education Key Laboratory for Earth System Modeling
- (Tsinghua University) (74110-71010025). Work partly carried out while WJL was a
- visiting student at the UoE supervised by SFBT.
- Review statement
- This paper was edited by Dr. Tilo Ziehn and reviewed by two anonymous referees.

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

**Figure 1**. Automatic tuning framework structure. Perturbed simulation results for each parameter are used for sensitivity analysis and determining the trust region size. Two key covariance metrics—observational error and model internal variation—help adjust parameter values in the objective function. The DFO-LS algorithm optimizes the parameters, and the post-processing module analyzes sensitivity, cost function results, and generates visualizations.

**Table 1**: Observations used for model evaluation, along with their target values and associated uncertainties.

| Variables name | Description                                  | Classifications       | Target | Uncertainty |
|----------------|----------------------------------------------|-----------------------|--------|-------------|
| MSLP           | Mean sea level pressure (hPa);               | MSLP_NHX_DGM          | 277.52 | 22.85       |
|                |                                              | MSLP_TROPICSLAND_DGM  | 35.42  | 13.69       |
|                |                                              | MSLP_TROPICSOCEAN_DGM | 187.34 | 1.04        |
| T500           |                                              | TEMP@500_NHX          | 251.42 | 0.12        |
|                | Temperature at                               | TEMP@500_SHX          | 249.38 | 0.56        |
|                | 500hPa (K)                                   | TEMP@500_TROPICSLAND  | 266.27 | 0.27        |
|                |                                              | TEMP@500_TROPICSOCEAN | 266.60 | 0.23        |
| RH500          | Relative<br>humidity at<br>500hPa (%)        | RH@500_NHX            | 52.75  | 7.04        |
|                |                                              | RH@500_SHX            | 51.05  | 4.79        |
|                |                                              | RH@500_TROPICSLAND    | 40.36  | 6.67        |
|                |                                              | RH@500_TROPICSOCEAN   | 32.57  | 3.01        |
| NETFLUX        | Net heat flux at top of atmosphere $(W/m^2)$ | netflux_GLOBAL        | 0.98   | 0.15        |

| Outgoing long                                                                                                                                                                                                                                                                                                                                                                                                                                                                                                                                                                                                                                                                                                                                                                                                                                                                                                                                                                                                                                                                                                                                                                                                                                                                                                                                                                                                                                                                                                                                                                                                                                                                                                                                                                                                                                                                                                                                                                                                                                                                                                                  |         |
|--------------------------------------------------------------------------------------------------------------------------------------------------------------------------------------------------------------------------------------------------------------------------------------------------------------------------------------------------------------------------------------------------------------------------------------------------------------------------------------------------------------------------------------------------------------------------------------------------------------------------------------------------------------------------------------------------------------------------------------------------------------------------------------------------------------------------------------------------------------------------------------------------------------------------------------------------------------------------------------------------------------------------------------------------------------------------------------------------------------------------------------------------------------------------------------------------------------------------------------------------------------------------------------------------------------------------------------------------------------------------------------------------------------------------------------------------------------------------------------------------------------------------------------------------------------------------------------------------------------------------------------------------------------------------------------------------------------------------------------------------------------------------------------------------------------------------------------------------------------------------------------------------------------------------------------------------------------------------------------------------------------------------------------------------------------------------------------------------------------------------------|---------|
| of atmosphere (W/m²) OLR_TROPICSCEAN 255.09 OUtgoing long OLRC_NHX 247.71 wave clearsky OLRC_SHX 243.59 OLRC flux at top of atmosphere (W/m²) OLRC_TROPICSOCEAN 290.21 Outgoing RSR_NHX 100.91 shortwave flux RSR_SHX 107.55 atmosphere (W/m²) RSR_TROPICSOCEAN 290.21 Outgoing RSR_TROPICSLAND 116.04 (W/m²) RSR_TROPICSOCEAN 86.92 Outgoing RSR_TROPICSOCEAN 86.92 Outgoing RSR_NHX 57.98 shortwave clearsky flux at RSRC_SHX 53.65 RSRC RSRC 55.0                                                                                                                                                                                                                                                                                                                                                                                                                                                                                                                                                                                                                                                                                                                                                                                                                                                                                                                                                                                                                                                                                                                                                                                                                                                                                                                                                                                                                                                                                                                                                                                                                                                                           |         |
| of atmosphere (W/m²) OLR_TROPICSLAND OLR_TROPICSOCEAN OUtgoing long OLRC_NHX VAY-71 Wave clearsky OLRC_SHX OLRC_SHX OLRC_SHX OLRC_SHX OLRC_TROPICSLAND OLRC_TROPICSLAND OLRC_TROPICSOCEAN OUtgoing RSR_NHX OUtgoing RSR_NHX RSR_SHX RSR At top of atmosphere (W/m²) RSR_TROPICSLAND ATMOSPHERE (W/m²) RSR_TROPICSLAND OUtgoing RSR_TROPICSLAND RSR_TROPICSLAND OUtgoing RSR_TROPICSOCEAN RSR_TROPICSOCEAN RSR_SHX ST-SHX | OLR     |
| OUTGOING LONG OUTC_NHX 247.71  wave clearsky OLRC_SHX 243.59 OLRC_SHX 243.59 OLRC_TROPICSLAND 288.64 (W/m²) OLRC_TROPICSOCEAN 290.21  Outgoing RSR_NHX 100.91 shortwave flux RSR_SHX 107.55 RSR at top of atmosphere (W/m²) RSR_TROPICSOCEAN 290.21  Outgoing RSR_NHX 100.91 shortwave flux RSR_SHX 107.55 ATMOSPHERE RSR_TROPICSLAND 116.04 (W/m²) RSR_TROPICSOCEAN 86.92 Outgoing RSR_TROPICSOCEAN 86.92 Outgoing RSRC_NHX 57.98 shortwave clearsky flux at RSRC_SHX 53.65 RSRC SRC_SHX 53.65                                                                                                                                                                                                                                                                                                                                                                                                                                                                                                                                                                                                                                                                                                                                                                                                                                                                                                                                                                                                                                                                                                                                                                                                                                                                                                                                                                                                                                                                                                                                                                                                                                |         |
| Wave clearsky  OLRC_SHX 243.59  OLRC_SHX 243.59  OLRC_TROPICSLAND 288.64  (W/m²) OLRC_TROPICSOCEAN 290.21  Outgoing RSR_NHX 100.91  shortwave flux RSR_SHX 107.55  at top of atmosphere RSR_TROPICSLAND 116.04  (W/m²) RSR_TROPICSOCEAN 86.92  Outgoing RSR_TROPICSOCEAN 86.92  Outgoing RSR_TROPICSOCEAN 86.92  Outgoing RSR_TROPICSOCEAN 86.92  Outgoing RSRC_NHX 57.98  shortwave clearsky flux at RSRC_SHX 53.65  RSRC 5.0                                                                                                                                                                                                                                                                                                                                                                                                                                                                                                                                                                                                                                                                                                                                                                                                                                                                                                                                                                                                                                                                                                                                                                                                                                                                                                                                                                                                                                                                                                                                                                                                                                                                                                 |         |
| OLRC_SHX 243.39  OLRC_SHX 243.39  At top of atmosphere (W/m²) OLRC_TROPICSOCEAN 290.21  Outgoing RSR_NHX 100.91  shortwave flux RSR_SHX 107.55  RSR at top of atmosphere (W/m²) RSR_TROPICSLAND 116.04  (W/m²) RSR_TROPICSOCEAN 86.92  Outgoing RSR_TROPICSOCEAN 86.92  Outgoing RSR_TROPICSOCEAN 86.92  Outgoing RSRC_NHX 57.98  shortwave clearsky flux at RSRC_SHX 53.65  RSRC 5.0                                                                                                                                                                                                                                                                                                                                                                                                                                                                                                                                                                                                                                                                                                                                                                                                                                                                                                                                                                                                                                                                                                                                                                                                                                                                                                                                                                                                                                                                                                                                                                                                                                                                                                                                          | OLRC    |
| OLRC_TROPICSLAND 288.64  (W/m²) OLRC_TROPICSOCEAN 290.21  Outgoing RSR_NHX 100.91  shortwave flux RSR_SHX 107.55  RSR at top of atmosphere RSR_TROPICSLAND 116.04  (W/m²) RSR_TROPICSOCEAN 86.92  Outgoing RSR_TROPICSOCEAN 86.92  Outgoing RSRC_NHX 57.98  shortwave clearsky flux at RSRC_SHX 53.65  RSRC TROPICSOCEAN 55.0                                                                                                                                                                                                                                                                                                                                                                                                                                                                                                                                                                                                                                                                                                                                                                                                                                                                                                                                                                                                                                                                                                                                                                                                                                                                                                                                                                                                                                                                                                                                                                                                                                                                                                                                                                                                  |         |
| Outgoing RSR_NHX 100.91 shortwave flux RSR_SHX 107.55 RSR at top of atmosphere (W/m²) RSR_TROPICSOCEAN 86.92 Outgoing RSR_TROPICSOCEAN 86.92 Outgoing RSRC_NHX 57.98 shortwave clearsky flux at RSRC_SHX 53.65 RSRC TROPICSOCEAN 85.95 RSRC_SHX 53.65                                                                                                                                                                                                                                                                                                                                                                                                                                                                                                                                                                                                                                                                                                                                                                                                                                                                                                                                                                                                                                                                                                                                                                                                                                                                                                                                                                                                                                                                                                                                                                                                                                                                                                                                                                                                                                                                          |         |
| shortwave flux  RSR_SHX 107.55  RSR at top of atmosphere (W/m²) RSR_TROPICSLAND 116.04 (W/m²) RSR_TROPICSOCEAN 86.92  Outgoing RSRC_NHX 57.98 shortwave clearsky flux at RSRC_SHX 53.65  RSRC top of 5.0                                                                                                                                                                                                                                                                                                                                                                                                                                                                                                                                                                                                                                                                                                                                                                                                                                                                                                                                                                                                                                                                                                                                                                                                                                                                                                                                                                                                                                                                                                                                                                                                                                                                                                                                                                                                                                                                                                                       |         |
| RSR at top of atmosphere RSR_TROPICSLAND 116.04  (W/m²) RSR_TROPICSOCEAN 86.92  Outgoing RSRC_NHX 57.98 shortwave clearsky flux at RSRC_SHX 53.65  RSRC top of 5.0                                                                                                                                                                                                                                                                                                                                                                                                                                                                                                                                                                                                                                                                                                                                                                                                                                                                                                                                                                                                                                                                                                                                                                                                                                                                                                                                                                                                                                                                                                                                                                                                                                                                                                                                                                                                                                                                                                                                                             | RSR     |
| RSR_TROPICSLAND 116.04 (W/m²) RSR_TROPICSOCEAN 86.92  Outgoing RSRC_NHX 57.98 shortwave clearsky flux at RSRC_SHX 53.65  top of 5.0                                                                                                                                                                                                                                                                                                                                                                                                                                                                                                                                                                                                                                                                                                                                                                                                                                                                                                                                                                                                                                                                                                                                                                                                                                                                                                                                                                                                                                                                                                                                                                                                                                                                                                                                                                                                                                                                                                                                                                                            |         |
| (W/m²) RSR_TROPICSOCEAN 86.92  Outgoing RSRC_NHX 57.98 shortwave clearsky flux at RSRC_SHX 53.65  RSRC top of 5.0                                                                                                                                                                                                                                                                                                                                                                                                                                                                                                                                                                                                                                                                                                                                                                                                                                                                                                                                                                                                                                                                                                                                                                                                                                                                                                                                                                                                                                                                                                                                                                                                                                                                                                                                                                                                                                                                                                                                                                                                              |         |
| shortwave clearsky flux at RSRC top of                                                                                                                                                                                                                                                                                                                                                                                                                                                                                                                                                                                                                                                                                                                                                                                                                                                                                                                                                                                                                                                                                                                                                                                                                                                                                                                                                                                                                                                                                                                                                                                                                                                                                                                                                                                                                                                                                                                                                                                                                                                                                         |         |
| RSRC clearsky flux at RSRC_SHX 53.65 5.0                                                                                                                                                                                                                                                                                                                                                                                                                                                                                                                                                                                                                                                                                                                                                                                                                                                                                                                                                                                                                                                                                                                                                                                                                                                                                                                                                                                                                                                                                                                                                                                                                                                                                                                                                                                                                                                                                                                                                                                                                                                                                       | RSRC    |
| ton                                                                                                                                                                                                                                                                                                                                                                                                                                                                                                                                                                                                                                                                                                                                                                                                                                                                                                                                                                                                                                                                                                                                                                                                                                                                                                                                                                                                                                                                                                                                                                                                                                                                                                                                                                                                                                                                                                                                                                                                                                                                                                                            |         |
| atmosphere RSRC_TROPICSLAND 75.67                                                                                                                                                                                                                                                                                                                                                                                                                                                                                                                                                                                                                                                                                                                                                                                                                                                                                                                                                                                                                                                                                                                                                                                                                                                                                                                                                                                                                                                                                                                                                                                                                                                                                                                                                                                                                                                                                                                                                                                                                                                                                              |         |
| (W/m <sup>2</sup> ) RSRC_TROPICSOCEAN 42.42                                                                                                                                                                                                                                                                                                                                                                                                                                                                                                                                                                                                                                                                                                                                                                                                                                                                                                                                                                                                                                                                                                                                                                                                                                                                                                                                                                                                                                                                                                                                                                                                                                                                                                                                                                                                                                                                                                                                                                                                                                                                                    |         |
| Land Lprecip_NHX 1.60e-8 0.35e-9                                                                                                                                                                                                                                                                                                                                                                                                                                                                                                                                                                                                                                                                                                                                                                                                                                                                                                                                                                                                                                                                                                                                                                                                                                                                                                                                                                                                                                                                                                                                                                                                                                                                                                                                                                                                                                                                                                                                                                                                                                                                                               | Lprecip |
| Lprecip precipitation Lprecip_SHX 1.42e-8 4.29e-9                                                                                                                                                                                                                                                                                                                                                                                                                                                                                                                                                                                                                                                                                                                                                                                                                                                                                                                                                                                                                                                                                                                                                                                                                                                                                                                                                                                                                                                                                                                                                                                                                                                                                                                                                                                                                                                                                                                                                                                                                                                                              |         |
| (m/s) Lprecip_TROPICSLAND 4.47e-8 0.37e-9                                                                                                                                                                                                                                                                                                                                                                                                                                                                                                                                                                                                                                                                                                                                                                                                                                                                                                                                                                                                                                                                                                                                                                                                                                                                                                                                                                                                                                                                                                                                                                                                                                                                                                                                                                                                                                                                                                                                                                                                                                                                                      |         |
| LAT_NHX 275.72- 0.06                                                                                                                                                                                                                                                                                                                                                                                                                                                                                                                                                                                                                                                                                                                                                                                                                                                                                                                                                                                                                                                                                                                                                                                                                                                                                                                                                                                                                                                                                                                                                                                                                                                                                                                                                                                                                                                                                                                                                                                                                                                                                                           | T2M     |
| T2M Temperature at 2 meters (K) LAT_SHX 280.08 0.49                                                                                                                                                                                                                                                                                                                                                                                                                                                                                                                                                                                                                                                                                                                                                                                                                                                                                                                                                                                                                                                                                                                                                                                                                                                                                                                                                                                                                                                                                                                                                                                                                                                                                                                                                                                                                                                                                                                                                                                                                                                                            |         |
| LAT_TROPICSLAND 297.10 0.31                                                                                                                                                                                                                                                                                                                                                                                                                                                                                                                                                                                                                                                                                                                                                                                                                                                                                                                                                                                                                                                                                                                                                                                                                                                                                                                                                                                                                                                                                                                                                                                                                                                                                                                                                                                                                                                                                                                                                                                                                                                                                                    |         |

**Table 2**: Summary of tunable parameters in GAMIL3, including their default values and plausible ranges.

| Parameters | Description (units if applicable)                   | Range        | Default |
|------------|-----------------------------------------------------|--------------|---------|
|            | Description (units if applicable)                   |              | Values  |
| c0_conv    | Precipitation efficiency for deep convection        | 1.e-4-5.e-3  | 1.e-3   |
| rhcrit     | Threshold value for RH for deep convection          | 0.65-0.95    | 0.85    |
| capelmt    | Threshold value for cape for deep convection (J/kg) | 20-200       | 70      |
| alfa       | Initial deep convection cloud downdraft mass flux   | 0.05-0.6     | 0.2     |
| ke         | Evaporation efficiency of deep convection           | 1.e-6-1.5e-5 | 9.e-6   |
|            | precipitation ()                                    | 1.6-0-1.36-3 |         |
| c0         | Rain water autoconversion coefficient (1/m)         | 3.e-5-2.e-4  | 5.e-5   |
| cmftau     | Characteristic adjustment time scale (s)            | 1800-14400   | 4800    |
| rhminl     | Threshold RH for low stable clouds                  | 0.8-0.99     | 0.95    |

| rhminh  | Threshold RH for high stable clouds                                              | 0.4-0.99    | 0.5   |
|---------|----------------------------------------------------------------------------------|-------------|-------|
| dthdpmn | Most stable lapse rate below 750hPa, stability trigger for stratus clouds (K/mb) | -0.150.05   | -0.08 |
| sh1     | Amplification factor (shallow convective cloud fraction)                         | 0.0-1.0     | 0.04  |
| sh2     | Scale factor for shallow convective mass flux                                    | 10-1000     | 500   |
| dp1     | Amplification factor (deep convective cloud fraction)                            | 0.0-1.0     | 0.1   |
| dp2     | Scale factor for deep convective mass flux                                       | 10-1000     | 500   |
| ccrit   | Minimum allowable sqrt(TKE)/wstar                                                | 0.0-1.0     | 0.5   |
| dzmin   | Minimum cloud depth to precipitate (m)                                           | 0.0-100.0   | 0.0   |
| Dcs     | Autoconversion size threshold for ice to snow (m)                                | 1.e-5-1.e-3 | 2.e-4 |
| ecr     | Collection efficiency cloud droplets/rain                                        | 0.5-2.0     | 1.0   |
| ai      | Fall speed parameter for stratiform cloud ice (1/s)                              | 500-1500    | 700   |
| qcvar   | Inverse relative variance of subgrid scale cloud water                           | 0.1-2.0     | 1.0   |

**Figure 2**. All experiments conducted in this study, including the AMIP2011 optimization runs for 10- and 20-parameter cases, the AMIP2005-2014 simulations using the optimized parameter sets, and the 30-year piControl simulations. Note that piControl simulations were not performed for the sensitivity experiments that varied the initial values of the 10 parameters (shown in brown).

**Figure 3**. Normalized values of tuning parameters for the default and all five optimized cases (a); changes in the cost function values over numbers of evaluations for the two main optimized cases (b) and the three sensitivity experiment cases (c). The vertical solid lines indicate the 11 and 21 runs from the initial perturbation phase, while vertical dashed lines mark the iterations at which the cost function reach its minimum.

**Figure 4**. Z values for the AMIP2011 (a) and AMIP2005-2014 (b) simulations. Solid and hollow dots represent tuning with 10 and 20 parameters, respectively. Blue dots indicate improved performance, while red dots show deterioration. The black dashed line at Z = 0 separates improved from non-improved variables.

**Figure 5**. AMIP2011 results (dots) and time series (lines) for three cases for: T500 (a), RH500 (b), OLR (c), OLRC (d), RSR (e), RSRC (f), T2M (g), Lprecip (h), MSLP (i) and NETFLUX (j). The cases include the default case (orange lines and dots), 10-parameter case (blue lines and dots), and 20-parameter case (red lines and dots). The black lines and shadings represent the observations and their associated uncertainties.

**Figure 6**. Taylor-diagram showing all variables for three cases in 2011 (a) and the AMIP2005-2014 simulations (b). Shown are default case (yellow), 10-parameter case (blue), and 20-parameter case (red).

**Figure 7**. Meridional distributions of the annual mean bias between three cases and observations for: T500 (a), RH500 (b), OLR (c), OLRC (d), RSR (e), RSRC (f), T2M (g), Lprecip (h) and MSLP (i) from the AMIP2005-2014 simulations. Shown are default case (orange), 10-parameter case (blue), and 20-parameter case (red).

**Figure 8**. Normalized Jacobian for all 20 parameters, with values normalized by the total covariance metrics. The x-axis shows the parameter names, while the y-axis represents the variables. Black parameters are used in the 10-parameter case, and green ones are added in the 20-parameter case. Red and blue indicate positive and negative effects, respectively, with darker shades showing greater impact.

**Figure 9**. Latitude-pressure anomaly distributions relative to the default case for relative humidity (a, b), cloud fraction (c, d), and temperature (e, f) from AMIP2005-2014 simulations: 10-parameter case (a, c, e) and 20-parameter case (b, d, f).

**Figure 10**. Results from the 30-year piControl simulation for NETFLUX (a), RSR (b) and OLR (c) radiation, mean volume-averaged ocean temperature (d), and T2M in the default (orange), 10-parameter (blue), and 20-parameter cases (red) cases.

**Figure 11**. Sea surface temperature biases relative to observations (HadISST; Rayner et al., 2003) from the last 15 years of piControl simulations for the default case (a, b, c) and two optimized cases (d-i).

**Figure 12**. Distribution of shortwave (a, b) and longwave (c, d) cloud forcing differences between the two optimized cases and the default case.

**Figure 13**. Similar as Fig. 7, but showing the range of Jacobians calculated from the optimized parameter set across four cases: the original optimized case and three sensitivity cases.