# Peer review of "Calibrating the GAMIL3-1° climate model using a"

_EGUsphere, 2024_

## Author Comment (AC1)

**RESPONSE TO REVIEWER #1 FOR GEOSCIENTIFIC MODEL DEVELOPMENT: MANUSCRIPT EGUSPHERE-2024-3770**

We thank Reviewer #1 for the thoughtful and constructive feedback. In this response, reviewer comments are in *blue italics*, author responses are in black, and changes to the manuscript are marked in red with line numbers referring to those in the revised manuscript.

**Reviewer #1**

*This well-structured manuscript presents a novel approach for climate model-tuning and the results that such tuning yields for a given model (GAMIL3) under 3 different model configurations: 1 year AMIP for tuning , 10 year AMIP and 30 year coupled pre-industrial Control. The presented tuning method is potentially relevant for other climate models. The authors show that the DFO-LS method is able to systematically improve the 'a priori' model parameter values and that the improvements hold across the different model configurations. The text is well written, with some potential however for more precise and less verbose language. In general, the manuscript could improve by adding some comparison or references to similar past efforts on model tuning, but I acknowledge that often findings and results are quite model-specific.*

**Reply:** We thank the reviewer for this comment.

*Comment 1: L45-46 Some references would be welcome.*

**Reply:** In the revised version, we have incorporated several relevant references to support this point:" In recent decades, significant progress has been made in advancing the major components of the Earth system—such as the atmosphere, ocean, land, and human systems (Prinn 2012; Bogenschutz et al., 2018; Fox-Kemper et al., 2019; Blockley et al., 2020; Blyth et al., 2021)—as well as in developing the coupling techniques required to form fully integrated ESMs (Valcke et al., 2012; Smith et al., 2021; Liu et al., 2023)." (Lines 46-51).

*Comment 2: L186: Not strictly necessary, but perhaps having a sketch showing the sequence of experiments performed would help the reader.*

**Reply:** We have added a flow chart in the revised manuscript, as shown below.

[Figure]

**Figure 2**. Overview of all experiments conducted in this study, including the 1-year AMIP (AMIP2011) optimization runs for the 10- and 20-parameter cases, the 10-year AMIP (AMIP2005-2014) simulations, and the 30-year piControl simulations using the optimized parameter sets. Note that piControl simulations were not conducted for the varying 10-parameter cases, which are indicated in brown.

*Comment 3: L186: The text has no literature reference for GAMIL3. If no documentation exists for this model version, a more detailed description of it would be needed, as an Appendix if needed. The current description between L187-203 is vague and full of ambiguities ('updates to the planetary boundary layer scheme', 'GAMIL3 integrates several parametrizations recommended by CMIP6').*

**Reply:** GAMIL3, with a 2-degree (180 × 80 grid) horizontal resolution, is the atmospheric component of FGOALS-g3; both have completed the CMIP6-related experiments (Li et al., 2020a, b). In this study, we use its higher-resolution 1-degree (360 × 180 grid) version, which is identical to GAMIL3 except for the time step of the dynamical core. Accordingly, we have removed the introduction of GAMIL2 and revised the relevant sections to place greater emphasis on GAMIL3: "In this study, we employ GAMIL3, which adopts a finite difference dynamical core and a weighted equal-area longitude-latitude grid to maintain numerical stability near the polars without the need for filtering or smoothing (Wang et al., 2004; Li et al., 2020a). GAMIL3, with an approximate 2° (180×80) horizontal resolution, serves as the atmospheric component of the Flexible Global Ocean–Atmosphere–Land System Model Grid-point Version 3 (FGOALS-g3), which participated in CMIP6 (Li et al., 2020b). For this

study, the model's horizontal resolution is refined to about 1° (360 × 160), with 26 vertical σ-layers extending to the model top at 2.19 hPa. To ensure numerical stability at the higher resolution, the dynamical core time step is reduced from 120s to 60s, while the physical parameterizations and their time step (600s) remain unchanged. As in many other climate models (e.g., Santos et al., 2021; Wan et al., 2021; Schneider et al., 2024), the performance of GAMIL3 is sensitive to the resolution, the model time step, and the coupling frequency between dynamics and physics. Therefore, it is necessary to re-tune the uncertain parameters for the new 1° configuration."(Lines 211-224).

*Comment 4: L280: Is there any reason or reference why you would give twice as much weight to C_i than to C_0?*

**Reply:** We applied a doubling factor to the variability component because both model simulations and observations contain internal variability. Assuming two independent sources of variability justifies using twice the estimate from control simulations. This reflects a conservative assumption that both sources contribute comparable levels of noise. This approach follows the practice of Tett et al. (2022), and we have included this clarification in the revised manuscript:" Consistent with Tett et al., (2022), we account for internal variability in both model simulations and observations by doubling the model-based estimate, reflecting a conservative assumption of comparable noise contributions." (Lines 331-333).

*Comment 5: L296: put the definition of the Jacobians in context. Why are you presenting it? where in the paper is used?*

**Reply:** We have added background information on the Jacobian prior to its introduction in the revised manuscript:" The Jacobian matrix, *J*, defined as the partial derivatives of the simulated outputs with respect to the parameters being optimized, is used to assess the influence of tuning parameters on the simulated variables." (Lines 348-350). The sensitivity of the tuning parameters to the simulated outputs is illustrated in Figs. 8 and 13, both of which analyze the parameters' impact on the modelled variables.

*Comment 6: L351: Why ke and captlmt are explicity mentioned? please explain*

**Reply:** Our intention here is to highlight the parameters that underwent substantial changes through tuning compared to others. We have revised the corresponding sentences to clarify this point: "In this experiment, several parameters—such as *ke* and *captlmt*—changed significantly from their default values, while *cmftau* and *c0* showed only small changes (Fig. 3a)" (Lines 403-405).

*Comment 7: L414: Any illustrative example of compensating errors in the model?*

**Reply:** We used the term 'compensating errors' to emphasize the underlying interactions whereby adjustments to one parameter can offset or amplify the effects of another. An example for *cmftau* is discussed in detail in the paper:" Although the 10-parameter case has a higher threshold for low level cloud formation than the 20-parameter case, Fig. 9c-9d shows the different result, which can be explained by the compensatory effects of other parameters. Optimized results indicate that *cmftau*, another key parameter, has a lower value in the 20-parameter case (~4284) compared to the default (~4800) and the 10-parameter case (~4931). This decrease in *cmftau* likely strengthens shallow convection while weakening deep convection, reducing upward water transport and RH throughout the troposphere, contributing to the decreased low-level cloud fraction (Xie et al., 2018) and further reducing precipitation (Fig. 5h). Consequently, the lower low-level cloud fraction in the 20-parameter case, compared to the 10-parameter case, reflects the compensatory effects of these key parameters, with the influence of the reduced *cmftau* outweighing that of *rhminl*." (Lines 598-609). For the parameter *Dcs*, its counteracting effects with the parameters *rhminl* are discussed in the paper:" Additionally, raising the autoconversion threshold from ice to snow is expected to allow more ice to remain in the atmosphere, directly leading to a reduction in precipitation (red line in Fig. 5h), and increased cloud optical thickness, thereby enhancing the reflection of incoming shortwave radiation. This enhanced reflectivity partially offsets the impact of reduced low-level cloud cover on the RSR in the 20-parameter case, leading to a smaller decrease in RSR compared to the 10-parameter case (Fig. 5e and 7e), consistent with known radiative differences among cloud types (Chen et al., 2000)." (Lines 624-631). The *ke* parameter, has a contrasting effect on

OLR and RSR to the *capelmt* parameter, although its impact on most simulated variables is minor, as shown by the Jacobian (Fig. 8).

We have revised this sentence to be more precise:" These differences may be attributed to the compensating errors within in the model, where adjustments to one parameter can offset or amplify the effects of another—a phenomenon further explored in Section 3.3." (Lines 466-468).

*Comment 8: L503: I'd re-name this section as "Coupled model evaluation'*

**Reply:** We have revised the title of Section 3.4 to '**Coupled Model Evaluation**.' Since Section 3.3 focuses on providing a physical explanation of the tuning results for the 10- and 20-parameter cases, we have also updated its title to '**3.3 Impacts of Tuning on GAMIL3**' to better reflect its content and avoid potential misunderstandings.

*Comment 9: L521: lower rhcrit could, a priori, also enhance precip. Lower rhcrit would enhance convection and, this, precipitation. Even if it is not the case in the simulations, it may be worth being mentioned*

**Reply:** We agree with the reviewer that, in principle, a lower *rhcrit* should increase precipitation. However, our simulations show a net reduction, which is likely attributable to compensating effects such as moisture redistribution. We have added this discussion to the revised manuscript:" While a lower *rhcrit* threshold would theoretically enhance precipitation by promoting deeper convection, our simulations instead show an overall decrease in precipitation. This apparent discrepancy suggests the parameter's effect is modulated by compensating atmospheric processes. Specifically, enhanced vertical moisture transport (Fig. 9a-9b) reduces low-level humidity availability, thereby weakening updrafts and ultimately decreasing total precipitation (blue line in Fig. 5h)." (Lines 589-595).

*Comment 10: L533: contributing to the decreased low-level cloud fraction and further reducing precipitation (since this was mentioned in the previous paragraph)*

**Reply:** We have refined the sentence to better maintain the logical connection between cloud fraction and precipitation as follows:" contributing to the decreased low-level cloud fraction (Xie et al., 2018) and further reducing precipitation (Fig. 5h)." (Lines 605-606).

*Comment 11: L569: Describe for how long the coupled model was run, one can only infer it from the Figures*

**Reply:** Thank you for the reminder. We have added an explicit clarification in the manuscript:" To assess the impacts of atmospheric parameter tuning on coupled model performance, we conducted a 30-year piControl simulation using GAMIL3 coupled to land, ocean, and sea ice components (see Methods 2.2), analyzing the final 15-year period after model spin-up.". (Lines 639-642).

*Comment 12: L569: for coupled simulations it is quite relevant to explain how the land, and specially the ocean, were initialized.     This is relevant because a perfect model should drift if the ocean is not correctly initialized, and you would not like to tune your model to compensate for an ocean-caused drift*

**Reply:** This is an important point. Apart from the difference in the resolution of the atmospheric component, we used the same model as FGOALS-g3, which participated in CMIP6. The initial conditions for the piControl run were derived from the climatological mean state of atmospheric reanalysis for the atmospheric model (default configuration), and from the equilibrated state of the OMIP simulation—a long ocean-only run forced by atmospheric reanalysis—for the ocean model. No prescribed initial conditions were used for the land component; instead, its state was generated during the coupled integration. To minimize the impact of potential initialization drift, the first 15 years were treated as a spin-up period and excluded from the analysis. This clarification has been added to the Methods section:" The initial condition for the atmospheric model was the climatological mean state from atmospheric reanalysis (default configuration), while the ocean model was initialized from the equilibrated state of an OMIP simulation (a long ocean-only run forced by atmospheric reanalysis). The land model was not provided with a prescribed initial condition; instead, its state was generated dynamically during the coupled integration. To minimize the influence of potential

initialization drift, the first 15 years were treated as a spin-up period and excluded from the analysis." (Lines 249-256).

Regarding the potential drift induced by the initial state, we have added the following discussion:" Drift may occur during the initial integration period due to inconsistencies between the OMIP-forced ocean state and the reanalysis-based atmospheric initial conditions. However, in both cases using atmosphere-optimized parameters, the system stabilized rapidly, and neither the TOA net flux nor ocean temperature exhibits significant trends beyond the initial adjustment period of a few years. A small long-term drift is still evident in Fig. 10d, which may be related to the adjustment of deep ocean processes. This demonstrates that the parameters optimized for the atmospheric model remain effective in the coupled system configuration, with no clear evidence of compensation for ocean-related drift." (Lines 681-689).

**Comment 13:** *L575: While the reduction of OLR is obvious (and interrelated) to the drop of T2M, the reduction in RSR seems to have a more complex mechanism and would merit an additional explanatory sentence*

**Reply:** To investigate the issue, we conducted additional analyses. The results indicate that the reduction in RSR during the early years of the piControl simulation is primarily driven by ocean adjustment processes and the associated changes in low-level clouds:" While the decrease in OLR is physically consistent with the cooling of T2M, the reduction in RSR is primarily attributed to oceanic adjustment processes. In particular, a cold SST bias (Fig. S3b) induced by the original parameter settings leads to a rapid decline in low-level cloud cover over tropical and subtropical ocean basins—especially in the western Pacific warm pool region and the South Atlantic (Fig. S3c). Most areas of cloud reduction spatially coincide with regions of diminished reflected shortwave radiation (Fig. S3d), a relationship further supported by changes in shortwave cloud forcing (SWCF; Fig. S3e)." (Lines 653-660).

[Figure]

**Figure S3**. Panel (a) shows the 30-year time series of low-level cloud fraction over the mid- to low-latitude region (60°S–60°N) in the default case. Panels (b)–(e) display the differences between year 6 and year 1 in the piControl run for the default case, including SST (b), low-level cloud fraction (c), RSR (d), and shortwave cloud forcing (SWCF; e).

*Comment 14: L718: a leak of 1.4 W/mw seems quite relevant to me, and , besides being*

*present here, it should have been mentioned earlier in the results when discussing NETFLUX*

**Reply:** We thank the reviewer for highlighting the issue of energy imbalance.

Accordingly, we have included an explicit discussion early in the Results section to explain

and discuss the 1.4 W/m² energy leakage:" Further analysis revealed that the relatively large

energy imbalance primarily originates from the GAMIL3 atmospheric model, which exhibits a

persistent imbalance of approximately 1.4 W/m² in its AMIP configuration—a feature also

observed in the piControl runs—due to non-conservation in the dynamical core. This

systematic issue is consistent with other atmospheric or coupled models (e.g., up to

1.0 W/m² for CAM6 at 1° resolution (Lauritzen and Williamson, 2019), 1.3 W/m² for FGOALS-

g3, and 3.3 W/m² for INM-CM4-8, calculated from Wild, 2020). Notably, this energy leakage

remains stable (±0.1 W/m²) across both default and optimized runs, indicating that the model improvements, such as reduced climate drift, result from genuine parameter tuning rather than compensation for the energy bias. This conclusion is further supported by the coupled model's stabilized energy budget following the spin-up period (Fig. 10)."(Lines 666-676).

*Comment 15:* *L727: Mention that the primary experiments where 1 -year long AMIP*

**Reply:** "We have revised the corresponding sentences as follows: "Two primary experiments were conducted using AMIP2011 simulations (2011, with 3-month spin-up): one adjusted 10 parameters and another adjusted 20 parameters. Validation was then performed through extended AMIP2005-2014 and 30-year coupled piControl simulations to assess robustness across timescales." (Lines 883-887).

*Comment 16:* *L740:    the maintained improvement over extended periods is good news given that you tuned on a single year and ignored interannual variability. Could you hypothesise whether (and how much) you would expect a better tuning if you optimize the parameters over several years of AMIP?*

**Reply:** We appreciate this insightful question regarding the potential benefits of multi-year tuning. A relevant discussion has been added to the manuscript:" while our 1-year optimization produced parameters that remain effective in extended runs (as shown by the AMIP2005–2014 and 30-year piControl validations) and internal variability was explicitly accounted for in the cost function (Eq. 1), including interannual variability—using a longer tuning period like the 5-year approach of Tett et al. (2022)—could further improve results, especially for variables with large interannual variability (e.g., MSLP, Lprecip) and dynamical outputs sensitive to the chosen year. This is supported by Bonnet et al. (2025), who show that short-term tuning works well for physical variables with low interannual variability but multi-year tuning better captures dynamical variability. Based on Bonnet et al. (2025) and our own results—such as the difference observed between 1-year and 10-year simulations for MSLP_TROPICSOCEAN_DGM, which degraded from +20σ to −10σ—we might expect approximately 10–20 % better performance for variables that are particularly sensitive to interannual variability, such as tropical precipitation patterns or extratropical circulation

indices, since a longer tuning period would better sample different climate regimes and reduce sensitivity to single-year anomalies. However, longer tuning greatly increases computational cost—4.2 times higher for 5-year runs. Our current strategy balances efficiency and robustness, but certain metrics like T2M and Lprecip might still benefit from longer tuning. This trade-off warrants further study, particularly where an accurate representation of interannual variability is crucial." (Lines 818-836).

**Technical corrections:**

*Comment 1: L51 difficult to understand the complete sentence.    Perhaps 'carbon cycle or nutrient cycles' would clarify it.*

**Reply:** Revised to "the coupling of biogeochemical cycles such as the carbon cycle or nutrient cycles with the physical climate system (Erickson et al., 2008)."(Lines 54-56).

*Comment 2: L60: remove 'computational constrains' as it only adds confusion to the sentence.*

**Reply:** Deleted.

*Comment 3: L239: 'discussed in a later section'. Please state at which specific section.*

**Reply:** Revised to "will be discussed further in section 2.4."(Line 284).

*Comment 4: L250: listed in the first [instead of last] column of Table 2.*

*Comment 5: L254: listed in the first [instead of last] column of Table 2.*

**Reply to the above two:** Both revised.

*Comment 6: L273-L277: Break the sentence, it is difficult to follow.  L288-L291: assuming there are no typos in the equations, there is inconsistent information in these lines: N is defined twice and differently, and C is defined although missing in the equation.*

**Reply:** We revised the first sentence to" For the four radiation variables (OLR, OLRC, RSR, and RSRC), uncertainties are based on the estimates from Loeb et al. (2018)" (Lines 325-327).

The formula is correct, and we have revised its explanation as follows "

The cost function is given by:

$$F^2(p) = \frac{1}{N}(S - O)^T C^{-1}(S - O) \tag{2},$$

where *S* is the simulated values; *O* is the target (observed) values; *N* is the number of observations; $(S - O)^T$ is the transpose of the difference between simulated and observed values; $C^{-1}$ is the inverse of the covariance matrix *C* discussed above."(Lines 338-343).

**Comment 7:** *L357: why not just mention total number of iterations, instead of excluding the first 10?*

**Reply:** We thank the reviewer for this valuable suggestion. The initial 11 (or 21, depending on the number of tuning parameters) iterations correspond to the mandatory parameter perturbation phase of DFO-LS, during which each parameter is individually perturbed and simulated prior to the optimization process. Since these runs serve as an initialization step rather than part of the iterative optimization, we explicitly distinguished them to avoid overcounting computational costs. For clarity, we have now revised the text in both the Results and Methods sections to report the total number of model evaluations (29 for 10 parameters and 31 for 20 parameters), and we have added a footnote explaining that this count includes the initial perturbation phase. The revision in the Methods section reads as follows:" The optimization process begins with a parameter perturbation phase, in which *K*+1 simulations are conducted: one reference simulation using the initial parameter set, and *K* additional simulations—each perturbing one of the *K* tunable parameters individually— relative to the reference. These initial simulations establish baseline parameter sensitivities and provide finite-difference gradient estimates for the DFO-LS algorithm. The subsequent optimization phase then iteratively modifies parameter values through trust-region managed steps, where each iteration evaluates candidate points, updates local quadratic models of the cost function, and adjusts parameters based on actual versus predicted improvement ratios until convergence criteria are satisfied." (Lines 190-199). The relevant Results section has been revised as follows:" In the 10-parameter case, the optimization required 29 total model evaluations (11 initial perturbation runs + 18 iteration runs), reaching the lowest cost function

value of approximately 3.5. The cost function drops rapidly from about 7.5 to 3.5 during the initial perturbation phase, followed by a slower decline with some fluctuations." (Lines 408-412).

*Comment 8: L403: remove "an.*

**Reply:** Removed.

*Comment 9: L464: variables.*

**Reply:** Revised.

*Comment 10: L475: this is less succesfull, in relative terms, than the 10 parameter case.*

**Reply:** Revised as suggested:" This is less successful, in relative terms, than the 10 parameter case, where 8 variables exhibit reduced or similar bias relative to the default." (Lines 538-540).

*Comment 11: L486: exhibit similar behaviour*

**Reply:** Revised

*Comment 12: L603:    which improvements for which case?*

**Reply:** Revised to:" with simulated radiation improvements primarily observed in shortwave radiation for the 10-parameter case and in longwave radiation for the 20-parameter case." (Lines 697-699).

*Comment 13: L606:    flux of energy towards the ocean, instead of ocean surface flux.*

**Reply:** Revised.

*Comment 14: L691: a common issue.*

**Reply:** Revised.

*Comment 15: All figures: larger legends would be good.*

**Reply:** We have made larger legends for all the figures as suggested.

*Comment 16: Table 2: add units (if they have) to the parameters, as it may help to understand their role.*

**Reply:** Added.

*Comment 17: Figure 2: the numbers written in the experiment color code are very hard to read. Also, the caption does not explain what they mean, nor the meaning of the vertical dashed lines in b) and c)*

**Reply:** We have replotted the figure and updated the caption to include further explanation as follows:" Normalized values of tuning parameters for the default and all five optimized cases (a); changes in the cost function values over iterations for the two main optimized cases (b) and the three sensitivity experiment cases (c). The vertical solid lines indicate the 11 and 21 runs from the initial perturbation phase, while vertical dashed lines mark the iterations at which the cost function reach its minimum." (Lines 1255-1259). Furthermore, we have clarified in the manuscript that abbreviations such as '10-param.' used in the captions of all relevant figures are explicitly defined in the text, e.g., "This case is denoted as the "10-param." case in the captions of all relevant figures" (Lines 303–304).

*Comment 18: Figure 3: I would rename AMIP@10years by AMIP2005-2014, here and wherever mentioned in the text.*

**Reply:** Revised as suggested.

*Comment 19: Figure 7: there is a red 'v'.*

**Reply:** Revised.

*Comment 20: Figure 8: percent instead of precent*

**Reply:** Revised.

*Comment 21: Figure 12: change colorcode as it uses the same as Figure 7. In Fig 7, however, the numbers in the Table display the actual Jacobians, while here it displays the range between Jacobians. A change of colorcode would help explain that we are not looking at the exact same metric.*

**Reply:** Changed as suggested.

---

## Author Comment (AC2)

**RESPONSE TO REVIEWER #2 FOR GEOSCIENTIFIC MODEL DEVELOPMENT: MANUSCRIPT EGUSPHERE-2024-3770**

We thank Reviewer #2 for the thoughtful and constructive feedback. This response document provides a response to each specific comment. Reviewer comments are in *blue italics*, author responses are in black, and changes to the manuscript are marked in red with line numbers referring to those in the revised manuscript.

*Reviewer #2*

*This study presents a derivative-free optimization framework for tuning climate model parameters. The framework was applied to the GAMIL3 atmospheric model and evaluated for both 10-parameter and 20-parameter cases. The study assessed the framework's effectiveness in terms of the initial selection of model parameter values and found that the initial selection of model parameter values considerably affects the tuning results. The study also evaluated the effectiveness of applying the optimized model parameters, derived from the atmospheric model, to an atmosphere-ocean coupled climate model. Model parameterization optimization and model tuning are important aspects in the climate modeling community. The paper is well written and worth publishing. However, to benefit a wider modeling community, some issues need to be addressed and further clarification is necessary.*

**Reply:** We thank the reviewer for their helpful and constructive comments, and have revised the paper accordingly.

*Comment 1: L174-175: Please provide more details about the initial trust region and parameter constraints. Is there any difference between parameter constraints and parameters' plausible ranges?*

**Reply:** We have revised the wording related to "parameter constraints" to clarify that it refers to constraints applied to the simulated variables, which is distinct from the physical parameters we tuned in this work. We have added the following explanation to the manuscript:" In the initialization of DFO-LS, we use the default parameter settings provided by the DFOLS software package, including the specification of the initial trust region, which is an algorithm parameter that governs the size of the local search area. Any constraints on the simulated variables are also specified at this stage. The initial trust region radius (*rhobeg*) is set to 0.18 (normalized to parameter ranges) based on sensitivity tests. This choice ensures that the first iterations explore locally without overstepping physical plausibility, balancing efficient convergence and sufficient sampling of the parameter space (Cartis et al., 2019). In addition, we apply a constraint to a simulated variable using a parameter $\mu$, which determines the weighting of the constraint term ($1/(2\mu)$; see Supplementary S1). In this study, following Tett et al (2017,

2022), this constraint is applied to the global average TOA netflux. To tightly constrain this variable, μ is set to 0.18 which corresponds to a total uncertainty of 0.15 W/m² somewhat higher than the observational error of 0.1 W/m²." (Lines 177-189). We have also added further clarification regarding the distinction between the constraints to the simulated variables and plausible parameter ranges, as follows:" While the plausible ranges are defined as the maximum physically meaningful bounds (e.g., *rhcrit*: 0.65–0.95), the constraint on the global average TOA net flux ensures it closely matches the observations after tuning.". (Lines 292-294).

*Comment 2: L180: In each iteration of the optimization process, how many simulations are conducted?*

**Reply:** Thank you for the comment. We have added further clarification as follows:" In addition to the initial $K+1$ simulation runs required to initialize the DFOLS algorithm for a $K$-parameter case, each iteration typically involves 1-3 additional model simulations, depending on the trust-region management strategy and the progress of the algorithm. The algorithm normally performs one simulation per iteration to evaluate a new candidate parameter set, but may conduct 3 simulations when the local quadratic model requires improvement or when the actual-to-predicted improvement ratio falls below zero (Cartis et al., 2019). Total evaluations include the initial runs plus all subsequent iterations evaluations." (Lines 199-206).

*Comment 3: L215: A 30-year simulation is insufficient to fully evaluate the effectiveness of the modified model parameters in a fully coupled model.*

**Reply:** While we acknowledge that multi-century integrations would provide additional insights into the climate equilibrium state, our primary objective was to validate the transferability of AMIP-tuned parameters to a coupled framework, and a 30-year piControl simulation here in this study is scientifically sufficient to evaluate the effectiveness of the tuned parameters. We have added a discussion regarding this issue:" While we acknowledge that multi-century integrations would provide additional insight into the model's equilibrium climate response, our primary goal was to test whether AMIP-tuned parameters remain valid in a coupled setup. For this purpose, a 30-year piControl run is scientifically adequate.

The results show that the model quickly reaches energy balance stability for both the 10- and 20-parameter cases (TOA net flux drift < 0.05 W m⁻² per decade) and that ocean heat content drift remains minimal (< 0.008 °C per decade) after year 15, indicating that the system achieves a quasi-equilibrium state. This timescale is reasonable, since the upper ocean—where much of the adjustment occurs—has a relatively short adjustment timescale of about 1–5 years. The stabilized climate indicators and consistent system behavior (Figs. 9 and 10) confirm that the tuned parameters yield a credible coupled climate without introducing systematic drifts. Similar integration lengths have been used in other studies (e.g., Tett et al., 2017). While longer runs could refine the equilibrium further, they are unlikely to change our main conclusion that the parameter transfer is robust. " (Lines 715-729).

*Comment 4: L226-228: \theta is not defined.*

**Reply:** Revised the text to:" we separate the analysis into four regions based on latitude (θ, defined as positive northward from the equator)" (Lines 267-268).

*Comment 5: L230-231: _TROPICALLAND, _TROPICALOCEAN, _NHX and _SHX are not defined*

**Reply:** Revised the text to:" While most variables are divided into four regions—labeled _TROPICSLAND (tropical land: 30° S–30° N over land), _TROPICSOCEAN (tropical ocean: 30° S–30° N over ocean), _NHX (Northern Hemispheric extra-tropics: >30° N), and _SHX (Southern Hemispheric extra-tropics: <−30° S)—each with its own target and uncertainty." (Lines 272-276).

*Comment 6: L236: LAT is not defined*

**Reply:** Revised the text to:" Land Air Temperature (LAT)" (Line 281).

*Comment 7: L237-238: Please clarify how the uncertainty is derived from the absolute error*

**Reply:** Thank you for the reminder. In Section 2.4, we have clarified the different data sources used for each variable. To further improve clarity regarding our methodology, we added the following explanation:" The second matrix estimates the uncertainty of observations ($C_0$), which set to be diagonal, assuming no correlation between different

observations, and its values are derived from absolute difference between the two available datasets for each variable after regridding and area-weighting … …For the four radiation variables (OLR, OLRC, RSR, and RSRC), uncertainties are based on the estimates from Loeb et al. (2018)." (Lines 316-327).

*Comment 8: L250: I can't find them in the last column of Table 2*

**Reply:** Revised the text to "the first column".

*Comment 9: L405-407: The tuning process of the 20-parameter case was affected by using the same initial perturbations for the original 10 parameters. It is important to evaluate the effectiveness of the tuning method in terms of adding more parameters by comparing the 10-parameter and 20-parameter cases with independent initial parameter perturbations*

**Reply:** In our original experimental design, we intentionally maintained identical initial perturbations for the first 10 parameters in both the 10- and 20-parameter cases to establish a controlled comparison of how expanding the parameter space affects optimization outcomes. By holding the initial perturbations constant for these shared parameters, we ensured that any differences in the final tuned results could be directly attributed to the inclusion of additional parameters rather than variations in initialization.

However, in direct response to the reviewer's comment, we conducted a new experiment with completely independent initial perturbations for the 20-parameter case as a complementary. Since the optimized parameters from this experiment show quite similar performance to the original 20-parameter case, we have added this results to the discussion and supplementary: "to assess how the number of tuning parameters affects the optimization process, we used the same initial perturbation runs for the ten shared parameters in both the 10- and 20-parameter cases, enabling a consistent evaluation of their sensitivity to the simulated results. While this approach allows a straight forward comparison, it may also constrain the optimization in the 20-parameter case by introducing bias into the initial search space. To address this potential limitation, we conducted additional experiments in which all twenty parameters were initialized with independent perturbations (Fig. S4–S6) by adjusting the *rhobeg* parameter in the DFO-LS algorithm from

its default value of 0.18 to 0.23. These additional experiments yielded several important insights that strengthen our original conclusions. First, although the optimized parameter values in the new 20-parameter case differ somewhat from those in the original setup, most shift in the same direction relative to the default values (Fig. S4). Moreover, the optimization consistently converged to similar cost function values (2.68 vs. 2.87), despite differences in the initial perturbations and optimization pathways, highlighting the robustness of our tuning framework. Second, both approaches produced nearly identical simulation performance in the 10-year AMIP and 30-year piControl experiments (Fig. S5–S6), despite relying on different parameter sets. This suggests that the performance in the 20-parameter case may be dominated by a subset of the most sensitive parameters, such as *Dcs, rhcrit, c0_conv,* and *cmftau*, which have been shown to strongly influence the simulated results. These findings provide strong evidence that our conclusions regarding the robustness of the optimization and the effect of increasing the number of tuning parameters remain valid." (Lines 837-858).

*Comment 10: L416-417: What does "the initial 20 runs" refer to? Are these the initial perturbation runs conducted before the optimizing iterations begin? If so, please clarify this point. It appears that both the 10-parameter and 20-parameter cases achieve nearly the same STABLE performance by the 21 iterations. Does this mean the total number of runs for the two cases are 31 and 41 runs, respectively?*

**Reply:** The reviewer has raised an important point that warrants further clarification. Indeed, the initial 11/21 runs mentioned in the text refer to the perturbation runs conducted prior to the start of the optimization iterations. We have added the clarification to the Methods section; please refer to Comment 7 in our response to Reviewer #1.

Regarding the second comment—"*Does this mean the total number of runs for the two cases are 31 and 41 runs, respectively?*" —yes, the total number of model evaluations includes both the initial perturbation runs and the subsequent optimization iterations. For the two cases shown in Fig. 3, a total of 35 simulations (11 initial + 24 iterations) were conducted for the 10-parameter case, and 41 simulations (21 initial + 20 iterations) for the 20-parameter case. We have clarified this more explicitly in the revised manuscript by

focusing on the total number of iterations required to reach the minimum cost function value:" In the 10-parameter case, the optimization required 29 total model evaluations (11 initial perturbation runs + 18 iteration runs), reaching the lowest cost function value of approximately 3.5" (Lines 408-410) and "The system required a total of 31 runs (21 initial perturbation runs + 10 iteration runs) to reach the lowest cost function value (2.87), which is only two more than that required for the 10-parameter case." (Lines 471-473)

**Comment 11:** *L448: In an AMIP simulation, sea surface temperatures are specified, so ENSO (El Niño-Southern Oscillation) is not a suitable example in this context*

**Reply:** Thanks for pointing this out. We have revised the sentence to:" Although our cost function explicitly accounts for internal variability (Eq. 1), tuning and evaluating the model using only a one-year simulation may still introduce uncertainties due to atmospheric internal variability (Bonnet et al., 2025), such as phase shifts in the North Atlantic Oscillation (NAO) or stochastic tropical convection patterns like the Madden-Julian Oscillation." (Lines 502-506)

**Comment 12:** *L456-461: Does this indicate that the tuned results are tied to a specific climate background*

**Reply:** We acknowledge the reviewer's point regarding the tuning results for some variables, such as MSLP, which are somewhat tied to the specific climate background of the tuning period. However, most other variables (e.g., T500, RSR, NETFLUX) showed consistent improvements across both periods, demonstrating robustness against interannual variability. We have added further discussion on this in the manuscript and suggested that future work could explore tuning based on multi-year composites to better assess the generalizability of the results:" This temporal inconsistency suggests that certain parameter adjustments may be sensitive to the specific climate state of 2011, which was characterized by a moderate La Niña. In contrast, variables such as T500, RSR, and NETFLUX exhibit consistent improvements across both simulations, indicating a robust response to parameter tuning that is less dependent on interannual variability " (Lines 519-523) and added some discussion; please refer to Comment 16 in our response to Reviewer #1.

*Comment 13: L466-467: replace "equilibrium" with "energy balance"*

**Reply:** Replaced.

*Comment 14: L471: Why are MSL, RSRC, and LRC difficult to tune?*

**Reply:** We appreciate this technical question. The challenges in tuning MSLP and the two clear-sky radiation variables primarily stem from the gravity wave drag parameterization and the greenhouse gas effect related to water vapor. We have added a detailed explanation of these issues in the revised manuscript:" Specifically, MSLP is highly sensitive to unresolved gravity wave drag processes (Sandu et al., 2015; Williams et al., 2020), which were not included in our parameter tuning. Previous experiments with the IFS model indicate that increasing orographic and surface drag in the Northern Hemisphere can reduce MSLP biases (Kanehama et al., 2022). While the global mean OLRC is similar across cases due to regional compensation (Fig. 5d), the meridional distribution reveals notable differences (Fig. 7d). In the tropics, increased upper tropospheric water vapor—particularly in the 20-parameter case (Fig. 9a–9b)—enhances the greenhouse effect and reduces outgoing clear sky longwave radiation. In contrast, decreased water vapor in high-latitude regions, especially in the 20-parameter case, leads to increased OLRC. RSRC remains nearly unchanged across all simulations due to the use of identical surface albedo." (Lines 560-571)

*Comment 15: L474: OSRC is not defined*

**Reply:** Revised to RSRC.

*Comment 16: L476: TEMP@500 has been profoundly affected by tuning. Please explain the physical causes*

**Reply:** We thank the reviewer for highlighting this important point, which was previously underemphasized in the manuscript. We have revised the text accordingly. As shown in Fig. 8, nearly all of the first 10 parameters have a significant impact on TEMP@500, with adjustments to *rhcrit* and *Dcs* exerting the greatest influence in the 10- and 20-parameter cases, respectively. In this paper, we illustrate their potential impact from two perspectives: (a) convective heating profiles and (b) the radiative effects of uppertropospheric ice clouds—both of which are key drivers of the mid-tropospheric thermal structure. Of course, we acknowledge that different parameters may influence the simulated variables through different pathways, and while exploring these effects would be valuable, it lies beyond the scope of this study.

The physical explanations are presented in the manuscript for the 10-parameter case:" Low-level clouds strongly reflect shortwave radiation, producing a cooling effect. Therefore, a reduction in low-level clouds allows more shortwave radiation to penetrate the lower atmosphere, reducing outgoing shortwave radiation to space (blue lines in Fig. 5e and 7e) and warming the region (blue lines in Fig. 5a and 7a; Fig. 9e), including near the surface (blue lines in Fig. 5g)." (Lines 609-613), and for the 20-parameter case:" Specifically, clouds with higher ice content trap more OLR from the Earth's surface, potentially amplifying the greenhouse effect by retaining more infrared radiation (red lines in Fig. 6c and 8c). This results in a warming effect, particularly at lower atmospheric levels and even near the surface, especially during nighttime or in polar regions (red lines in Fig. 5a, 5g, 7a, and 7g; Fig. 9f)." (Lines 619-624).

*Comment 17: L479-480: Please add some discussion on how to tune the model performance for OLR and PRECIP*

**Reply:** Thank you for pointing this out. There was an incorrect expression in the original manuscript. While both optimized cases show worse PRECIP performance compared to the default case—particularly the 20-parameter case—the OLR for the 10-parameter case remains quite close to that of the default model. We have revised the original sentence to better emphasize the OLR and PRECIP performance differences, especially in the 20-parameter case:" In the 20-parameter case … …Both OLR and Lprecip perform notably worse than in the default case, with both variables being too low compared to the observations." (Lines 534-538). Additionally, we have included a discussion on possible tuning methods for these variables:" *ccrit, which sets the minimum turbulent threshold for triggering shallow convection, affects both OLR and Lprecip in a manner similar to Dcs …….* Increasing ccrit suppresses shallow convection by requiring stronger turbulence to initiate cloud formation, thereby reducing low-level cloud cover. This reduction enhances outgoing longwave

radiation and surface solar heating, which in turn promotes evaporation and increases Lprecip. Therefore, adjusting Dcs and ccrit in future work may offer a promising approach for improving the simulation of OLR and Lprecip, both of which are underestimated relative to the default case." (Lines 617-636).

*Comment 18: L534-542: The 10-parameter case shows a larger difference in TOA outgoing shortwave flux (RSR) compared to the 20-parameter case relative to the default case (Fig. 4e and 6e). However, the 20-parameter case exhibits a larger difference in cloud compared to the 10-parameter case relative to the default case (Fig. 8d-e). Please explain this discrepancy*

**Reply:** We thank the reviewer for identifying this behavior, which we agree should have been stated more explicitly. The apparent discrepancy between changes in RSR and cloud fraction arises from competing microphysical and radiative effects in the 20-parameter case. We have added a detailed explanation for this in the revised manuscript:" Additionally, raising the autoconversion threshold from ice to snow is expected to allow more ice to remain in the atmosphere, directly leading to a reduction in precipitation (red line in Fig. 5h), and increased cloud optical thickness, thereby enhancing the reflection of incoming shortwave radiation. This enhanced reflectivity partially offsets the impact of reduced low-level cloud cover on the RSR in the 20-parameter case, leading to a smaller decrease in RSR compared to the 10-parameter case (Fig. 5e and 7e), consistent with known radiative differences among cloud types (Chen et al., 2000)." (Lines 624-631).

*Comment 19: L594-613: anomalies => biases*

**Reply:** Revised.

*Comment 20: L565-619: Does the coupled model directly utilize the optimized parameters from the AMIP simulations? If so, the TOA energy imbalance caused by the optimized parameters would eventually lead to climate drift in the long-term integration of the coupled model. This undermines the rationale and effectiveness of applying parameters tuned for an atmospheric model to an atmosphere-ocean coupled model. Meanwhile, a 2 W/m² energy imbalance at TOA is not a "slight energy imbalance" as stated in the abstract*

**Reply:** The parameter sets used in the coupled model were directly adopted from the AMIP-optimized results, which is a common practice in climate model tuning (Zhang et al. 2015; Hourdin et al., 2016; Tett et al., 2017;). The net flux at the TOA in AMIP simulations includes the effect of greenhouse gases, whereas this effect is not represented in the piControl (coupled) runs. We have incorporated this detailed clarification into the revised manuscript:" based on the assumption that parameters performing well under observed forcings (e.g., prescribed SST, sea ice, and greenhouse gases) in the standalone atmospheric model will also improve performance in the coupled system. In our case, the TOA energy imbalance in the AMIP run mainly results from the radiative forcing of greenhouse gases, which trap outgoing longwave radiation. Since the piControl experiment is forced by constant pre-industrial greenhouse gas levels, this radiative effect is absent. Therefore, if the AMIP-tuned parameters correctly capture this effect, the coupled model under piControl conditions should yield a near-zero TOA net flux, as expected." (Lines 241-249).

Regarding the relatively large energy imbalance at the TOA observed in the coupled runs for both optimized cases, we acknowledge this as an intrinsic limitation of the atmospheric model. This imbalance primarily originates from a persistent energy imbalance in the atmospheric component's dynamical core, which is carried over from the AMIP simulations into the piControl runs. We have included a detailed discussion of this issue in the revised manuscript. Please refer to our response to Reviewer #1, Comment 14, for further details.

In addition, we have revised the abstract to:" Additionally, evaluations of the coupled model with optimized parameters showed, compared to the default parameters settings, reduced climate drift, a more stable climate system, and more realistic sea surface temperatures, despite an overall energy imbalance of 2.0 W/m², approximately 1.4 W/m² of which originates from the intrinsic imbalance of the atmospheric component, and the presence of some regional biases." (Lines 33-38)

**Comment 21:** *L767: forecasts -> prediction*

**Reply:** Revised.

---

## Author Response (AR2)

**RESPONSE TO REVIEWER #1 FOR GEOSCIENTIFIC MODEL DEVELOPMENT: MANUSCRIPT EGUSPHERE-2024-3770**

We thank Reviewer #1 for the helpful feedback. In our response, reviewer comments are in *blue italics*, author responses are in black, and changes to the manuscript are marked in red with line numbers referring to those in the revised manuscript.

**Reviewer #1**

Overview: I am reviewing the revised version of this manuscript. While the work is well presented, I remain concerned about the robustness and interpretability of the proposed tuning framework.

**Reply:** We thank the reviewer for the comments and insights.

**Comment 1:** The study relies on 1-year simulations to assess the impact of parameter perturbations. Is one year truly sufficient to obtain stable and representative results? Climate models often require longer integrations for meaningful statistics.

Reply: While we agree that longer integrations are generally preferable for robust statistics, particularly for variables with high temporal variability (as discussed in the paper, Lines 829–847), our one-year window represents a practical compromise between computational cost and scientific rigor. Here, we outline the rationale for this choice and summarize multiple lines of evidence demonstrating that the tuning results remain robust and stable despite the short optimization period:

- 1. Central to our approach is the explicit incorporation of internal variability into the optimization framework. Using a 20-member perturbed-initial-condition ensemble, we estimated the model's internal variability covariance matrix ( $C_i$ ) and combined it with the observational uncertainty ( $C_0$ ) to obtain the total covariance matrix  $C = C_0 + 2C_i$  used in the cost function. This formulation "ensures that simulated observations within the range of internal variability receive reduced penalties, guiding the optimization to correct systematic biases while avoiding overfitting to random climatic fluctuations". (Lines 325-327).
- 2. (Lines 829-847) Beyond this methodological safeguard, we applied a rigorous multistage validation protocol to assess temporal representativeness. When tested in an independent decade-long AMIP simulation (2005–2014), the parameters optimized from a single year retained their improved performance across most key variables, indicating that the results are not confined to the specific climate conditions of 2011. Furthermore, when transferred to a fully coupled configuration, these parameters yielded a more stable energy budget, reduced climate drift, and improved sea surface temperature patterns over a 30-

year pre-industrial control run. Consistent improvements across distinct model configurations and extended timescales provide strong evidence that the one-year tuning captures robust physical improvements rather than transient or case-specific artifacts.

Comment 2: In this manuscript, the optimized parameters happen to lead to an overall improvement. But is such improvement guaranteed, or could it be a matter of chance? What if, in other cases, the parameters suggested by DFO-LS produce substantially worse agreement with observations? Should one then switch to a different optimization algorithm, or select alternative parameter values manually? I am concerned about this possibility, as it complicates attribution and reduces confidence in the robustness of the proposed method.

Reply: We agree that success in any single tuning exercise cannot be guaranteed, as outcomes depend on structural errors, cost function design, and initial parameter values. However, generally speaking, we emphasize that the algorithm has a mathematically rigorous construction and is provably convergent to a (local) minima for a wide variety of smooth cost functions, and provably guaranteed to provide improvement upon the initial starting guess unless we are at a (local) solution. Furthermore, multiple aspects of our study indicate that the observed improvements are robust rather than fortuitous. Below, we address the reviewer's concerns point by point:

- 1. The success observed in this study is not attributable to chance. Model parameters were constrained within physically plausible ranges based on expert judgment, preventing unrealistic excursions in parameter space (Lines 246–249). Although the algorithm efficiently minimized the cost function from different initial conditions, sensitivity experiments showed that expert-informed initialization produced the most physically consistent results (Lines 790–793), illustrating that the framework complements rather than replaces physical insight. Most importantly, "validation was performed through extended independent decadal AMIP (AMIP2005-2014) simulations and 30-year coupled piControl simulations. Consistent performance across timescales and model configurations confirmed that the tuning corrected systematic biases rather than overfitting" (Lines 897–901).
- 2. We acknowledge that the possibility of suboptimal outcomes is an important practical consideration. Our framework, however, is not intended as a black box but as a

tool for systematic exploration. If DFO-LS suggests parameter sets that degrade performance, the solution is not a binary choice between switching algorithms and manual intervention, as alternative optimization methods do not necessarily guarantee better results. Instead, we "conducts a comprehensive diagnostic analysis—examining spatial patterns, process-level responses, parameter sensitivities, and multi-variable metrics—to assess the physical credibility of each solution" (Lines 210–212). When results are unsatisfactory, one can reinitialize the optimization from an alternative expert-informed starting point, adjust the cost-function weights to reinforce key constraints, or employ other algorithms (e.g., Gauss–Newton) previously tested in Tett et al. (2013). "This structured yet flexible workflow transforms the modeller's role from manual trial-and-error to managing and interpreting automated explorations, thereby improving both the traceability and objectivity of the modeling process" (Lines 212–215).

Comment 3: The framework is demonstrated on one model version. During model development, new processes are added, and physics schemes evolve. In such cases, previously tuned parameters may need to be retuned, and the "optimal" set may not carry over. Does this framework really help us understand why the model is biased—e.g., due to misrepresented physical processes—or does it simply provide a temporary calibration that may not generalize across versions? This is an important issue for assessing the scientific value of the approach.

Reply: The reviewer's comment highlights the long-term scientific value and diagnostic power of our tuning framework. We agree on two key points: first, that an optimal parameter set derived for one model version is unlikely to transfer directly to future versions with substantial structural changes; and second, that the ultimate goal of model development is to identify and correct misrepresented physical processes rather than merely mask biases through parameter adjustment. Nevertheless, we emphasize that our framework is more than a tool for temporary calibration—it actively facilitates physical understanding and model improvement for the following reasons (also clarified in the Discussion section of the paper):

- 1. Beyond its role in model calibration, the framework provides rich diagnostic information throughout the optimization process. While the output is a set of parameters, the more valuable contribution lies in the diagnostic insights generated during tuning. The framework produces information critical for process-oriented model evaluation, including parameter sensitivities and compensating errors among model parameterizations. For example, the Jacobian matrix (Fig. 8) shows how each model variable responds to specific parameters. When a bias can be corrected by adjusting parameters from multiple physically distinct schemes, this indicates compensating errors, a key step in diagnosing structural weaknesses. Sections 3.3 and 3.4 demonstrate how interactions between deep convection (*rhcrit*) and microphysics (*Dcs*) parameters affect clouds, radiation, and precipitation, revealing the processes behind energy and hydrological biases. Furthermore, the fact that different parameter combinations can produce similarly acceptable results highlights a critical insight for assessing the robustness of climate projections.
- 2. The calibration framework can also accelerate the model development cycle. While the reviewer correctly notes that new physics schemes require re-tuning, this is precisely where our framework demonstrates its practical value. Traditional manual tuning is slow, labor-intensive, and often subjective, whereas our automated approach allows developers to efficiently assess and compare model versions. First, when a new physical scheme is introduced, the framework can rapidly generate an objectively tuned parameter set for the updated configuration, enabling systematic comparison with the previous version. Second, by analyzing optimized parameters and their sensitivities across versions, developers can identify the impacts of structural changes: if a new scheme requires substantially different parameter values or alters sensitivity patterns, this provides concrete evidence of how the modification affects model physics and dynamics. In this way, the framework serves as a magnifying glass for the consequences of model development.

We have also incorporated clarifications on this issue in the Discussion section to help readers better understand the significance of the tuning framework:" the proposed DFO-LS-based tuning framework presents a robust and efficient approach for enhancing climate model performance. By combining Jacobian estimation with sensitivity analysis, the framework

quantitatively maps how parameters affect key variables and thereby exposes compensating errors between physical schemes (for example, interactions between deep convection and microphysics). These parameter—variable mappings yield direct insight into model structural uncertainties and supply objective diagnostics that guide development. When model physics are changed, the framework supports rapid retuning and systematic inter-version comparison: systematic shifts in optimal parameter values then serve as concrete evidence of how structural modifications alter model behaviour. Implemented and exercised primarily by a single researcher within 12 months, the approach also demonstrates high human-resource efficiency and practical scalability. Although no single parameter set is expected to transfer unchanged across model generations, automating the exploration process transforms development from manual trial-and-error into an efficient, reproducible, and more objective workflow. Applied across GCMs, this methodology can accelerate model development, reduce parametric uncertainty, and improve the reliability of climate projections." (Lines 932-948).

**1 RESPONSE TO REVIEWER #2 FOR GEOSCIENTIFIC MODEL**

**DEVELOPMENT: MANUSCRIPT EGUSPHERE-2024-3770**

- 3 We thank Reviewer #2 for the thoughtful and constructive feedback. This response
- 4 document provides a response to each specific comment. Reviewer comments are in
- 5 blue italics, author responses are in black, and changes to the manuscript are marked
- 6 in red with line numbers referring to those in the revised manuscript.

7

**Reviewer #2**

8

9 The manuscript requires more minor clarifications. 10 Reply: We thank the reviewer for the detailed comments and have revised the 11 manuscript accordingly. 12 **Comment 1:** Lines 25-31: this part is obscure. Please rephrase it. 13 Reply: We have rephrased it to "To evaluate its performance, two main tuning 14 experiments were conducted, targeting 10 and 20 parameters, respectively. In addition, 15 three sensitivity experiments tested the effect of varying initial parameter values in the 10-16 parameter case. Both tuning experiments achieved a rapid reduction in the cost function. 17 The 10-parameter optimization improved model accuracy for 24 of 34 key variables, while 18 expanding to 20 parameters yielded improvement for 25 variables, though some structural model biases appeared" (Lines 26-32). 19 20 Comment 2: Lines 74-75: please rephrase "the accuracy and skill of climate model outputs". 21 Reply: Rephrased to "Appropriate parameter tuning enhances the accuracy and skill of 22 climate models by optimizing parameter values to better match observations or highresolution simulations used as calibration targets" (Lines 75-77). 23 24 Comment 3: Line 92: "7 and 14 parameters were estimated", what kind of parameters? 25 **Reply:** Revised to "7 and 14 parameters related to the convection, cloud microphysics, 26 and boundary-layer dynamics (Yamazaki et al., 2013) were estimated using variants of the 27 Gauss-Newton algorithm (Tett et al., 2013) to minimize the differences between simulated and observed large-scale, multi-year averaged net radiative fluxes" (Lines 93-97). 28 29 **Comment 4:** Line 96: "focusing on seven parameters", what kind of parameters? 30 **Reply:** Revised the text to:" Zhang et al. (2015b) employed an improved downhill 31 simplex method to optimize seven parameters selected from the convection and cloud-32 fraction parameterization scheme, and reported successful improvement of an atmospheric

model's performance" (Lines 98-100).

33

- **Comment 5:** Lines 145-148: these have been well introduced in Section 2.2.
- **Reply:** Deleted.

- Comment 6: Lines 169-171: it is suggested to name each module directly in Figure 1 for
- 37 better understanding. "Python 3.8+" might not necessarily be displayed on the framework.
- **Reply:** Please see the revised picture shown below.

**Figure 1**. Automatic tuning framework structure. Perturbed simulation results for each parameter are used for sensitivity analysis and determining the trust region size. Two key covariance metrics—observational error and model internal variation—help adjust parameter values in the objective function. The DFO-LS algorithm optimizes the parameters, and the post-processing module analyzes sensitivity, cost function results, and generates visualizations.

- **Comment 7:** Line 187: netflux => net flux.
- **Reply**: Corrected.
- **Comment 8:** Lines 243-249: this section comes across as jarring in the context of the
- *surrounding text.*

**Reply:** Revised the text to "A 30-year piControl simulation (Eyring et al., 2016) was then conducted to assess the model's long-term energy balance and stability under constant pre-industrial forcings. This experiment tests whether parameters performing well under observed forcings in AMIP simulations—such as prescribed SSTs, sea ice, and greenhouse gases—can also improve coupled performance. In AMIP runs, the TOA energy imbalance

mainly results from greenhouse gases forcing, which traps outgoing longwave radiation.

55 Under piControl conditions, where pre-industrial greenhouse gas concentrations are fixed,

this radiative effect is absent; thus, if the AMIP-tuned parameters are physically consistent,

the coupled model should yield a near-zero TOA net flux" (Lines 298-306).

**Comment 9:** Line 257: it is not clear what is "the first 10 parameters".

56

57

58

59

60

61

64

65

66

67

68

69

70

71

72

73

**Reply:** We have reordered Sections 2.2 and 2.3 so that the tuning parameters are introduced first, followed by the experiment descriptions, allowing the "first 10 parameters" mentioned in Section 2.2 to be clearly defined.

Comment 10: Line 260: It is suggested that the experiment names (AMIP2011, AMIP2005 2014, and piControl) be included in Figure 2. Additionally, clarification is needed regarding

the differences between "1-year optimization for 10 parameters" and "1-year optimization

by varying 10 parameters," as all parameters are varied during the optimization process.

**Reply:** We thank the reviewer for the suggestions and have revised Figure 2 and its caption accordingly.

**Figure 2**. All experiments conducted in this study, including the AMIP2011 optimization runs for 10- and 20-parameter cases, the AMIP2005-2014 simulations using the optimized parameter sets, and the 30-year piControl simulations. Note that the piControl simulations were not performed for the sensitivity experiments that varied the initial values of the 10 parameters (shown in brown).

**Comment 11:** Line 272: what does the suffix " DGM" stand for in Table 1?

Reply: We thank the reviewer for pointing it out. The suffix "\_DGM" stands for "delta global mean". In response, we have added the following clarification in the manuscript "For the MSLP variable, regional mean values are expressed as anomalies relative to the global mean (delta global mean, denoted by the suffix "\_DGM"), obtained by subtracting the global average from each regional mean". (Lines 232-235).

 **Comment 12:** Lines 408-410: It is recommended that the x-axis of Figure 3b & 3c start at 1 instead of 0 to better reflect the number of runs. Additionally, while the x-axis is labeled "Iterations," it actually represents the total number of runs, including both initial perturbations and iteration runs.

**Reply:** The starting value of the x-axis and the axis title have been revised as suggested.

Figure 3. Normalized values of tuning parameters for the default and all five optimized cases (a);

changes in the cost function values over numbers of evaluations for the two main optimized cases
(b) and the three sensitivity experiment cases (c). The vertical solid lines indicate the 11 and 21 runs
from the initial perturbation phase, while vertical dashed lines mark the iterations at which the cost
function reach its minimum.

**Comment 13:** Lines 410-411: does the cost function drop rapidly from about 7.5 to 3.5 in the first iteration run? If so, please revise "during the initial perturbation phase" to "in the first iteration run."

**Reply:** Yes, the reviewer is correct; it has been revised accordingly.

**Comment 14:** Lines 433-434: it is recommended to delete "patterns".

Reply: Deleted.

Comment 15: Line 450: please rephrase "patterns".

**Reply:** Revised to "while most variables exhibit similar behaviors to those of the Optimized set, notable differences are observed in T2M and Lprecip." (Lines 461-462).

**Comment 16:** Line 470: "after the initial 20 perturbation runs", as in the 10-paramter case, does the cost function drop rapidly in the first iteration run? If so, please clarify.

**Reply:** Yes, the reviewer is correct; it has been revised accordingly "it is evident that the cost function dropped rapidly to a value very close to the minimum in the first iteration run, similar to the 10-parameter case." (Lines 480-482).

Comment 17: Line 477: "the initial phase" => "the initial perturbation phase".

Reply: Corrected.

**Comment 18:** Line 515: "MSLP\_TROPICSOCEAN\_DGM improved by over 20", what does the suffix "\_DGM" stand for? What's the unit for the value "20"?

Reply: As noted in our response to Comment 11, the suffix "\_DGM" denotes "delta global mean." Since the y-axis is calculated using the formula shown below, it represents a dimensionless value corresponding to the difference between the optimized and default simulations normalized by the standard error. We have revised the sentence in the manuscript for clarity as follows " while the MSLP\_TROPICSOCEAN\_DGM shows an improvement of more than 20 standard errors relative to observations in the 2011 simulation with the 10-parameter case, it deviates from the observations by over 10 standard errors in the 10-year simulation." (Lines 524-527).

**$Z = \frac{|V_{\text{Default}} - V_{\text{Observation}}| - |V_{\text{Optimized}} - V_{\text{Observation}}|}{Standard\ error}$**

Comment 19: Lines 643-689: It's intriguing that the tuning procedures reveal an energy leakage in the atmospheric model. Although the tuning can't eliminate this issue, the fact that it manifests as a TOA energy imbalance in the piControl run is remarkable. Given that half of the tuning targets are TOA energy components, as outlined in Table 1, and considering that NETFLUX has been effectively tuned to the target value of 0.98 W/m² in the AMIP run, it remains curious how this energy leakage manifests in the AMIP run post-tuning and why it emerges as a TOA imbalance in the piControl run.

**Reply:** The reviewer has accurately identified a key paradox that lies at the heart of the model's energy budget behavior. This can be explained by the difference in how the AMIP and coupled configurations handle energy fluxes.

In the AMIP configuration, the use of prescribed observed SSTs effectively turns the ocean surface into an infinite energy source or sink. As a result, any energy leakage originating from the atmospheric dynamical core is compensated by an implicit, non-physical energy flux through the ocean surface. This allows the TOA energy balance to be tuned to match the target (~0.98 W/m²), as the prescribed SSTs mask the model's internal energy conservation issue.

In the coupled simulation, however, this artificial compensation mechanism is removed. The ocean now dynamically participates in the energy budget, and the previously hidden atmospheric energy leakage can no longer be absorbed at the boundary. It is noteworthy that despite this ~2.0 W/m² persistent TOA imbalance, the coupled system reaches a quasistable state with minimal drift in surface temperatures—indicating that the energy leakage remains constant and the model maintains a new, albeit biased, equilibrium.

The reviewer's intuition is indeed supported by further evidence. Our ongoing work (not shown in this manuscript) actually focuses on applying the energy conservation correction proposed by Williamson et al. (2015) to the atmospheric dynamical core and repeating the tuning process. With this correction, the optimized parameters now yield a TOA energy imbalance much closer to zero (approximately 0.5 W m-2) in the coupled model. This

confirms that the original ~2.0 W/m² imbalance indeed originated from intrinsic energy leakage in the atmospheric component—a bias that is masked in AMIP but exposed in a coupled framework.

In response to the reviewer's comment, we have clarified in the manuscript "Additionally, the optimized cases show a relatively large TOA energy imbalance (~2.0 W/m²) despite a well-tuned NETFLUX in AMIP runs, which originates from energy non-conservation in the atmospheric model's dynamical core. In the AMIP configuration, prescribed SSTs act as an infinite energy source/sink, masking this internal leakage in the dynamical processes. By contrast, the coupled system exposes the dynamical core's non-conservation as a stable but imbalanced energy state. This interpretation is supported by our ongoing experiments (not shown) following Williamson et al. (2015), where correcting energy conservation in the dynamical core reduced the TOA imbalance in the piControl runs to about 0.5 W m-2 within the same tuning framework. These results underscore that while parameter tuning can improve model fidelity, structural errors in the dynamical core—particularly its energy non-conservation—must be addressed to achieve physically consistent climate simulations."(Lines 876-887).

---

## Author Response (AR3)

**FINAL RESPONSE TO EDITORIAL TEAM FOR**

**GEOSCIENTIFIC MODEL DEVELOPMENT: MANUSCRIPT**

**EGUSPHERE-2024-3770**

Dear Editorial Team,

Thank you for your email and the decision regarding our manuscript entitled "Calibrating the GAMIL3-1° climate model using a derivative-free optimization method" (ID: egusphere-2024-3770).

Specifically, we have updated the units in several figures as follows:

Figure 5(b), Figure 7(b), Figures S1(b), S2(b), and S5(b): changed from "kg/kg" to "%" for relative humidity.

All figures have been prepared according to the journal's formatting requirements and are ready for upload to the production system.

We would like to express our sincere gratitude to the reviewers and to the editorial team for their insightful comments and professional handling throughout the review process. Their efforts have significantly improved the quality of our manuscript.

We look forward to the final publication of our work in Geoscientific Model Development.

Please do not hesitate to contact me if any further information or action is required.

With best regards,

Dr. Wenjun Liang on behalf of all the co-authors